# Evaluation of Surgical Aid of Methylene Blue in Addition to Intraoperative Gamma Probe for Sentinel Lymph Node Extirpation in 116 Canine Mast Cell Tumors (2017–2022)

**DOI:** 10.3390/ani13111854

**Published:** 2023-06-02

**Authors:** Elisa Maria Gariboldi, Alessandra Ubiali, Lavinia Elena Chiti, Roberta Ferrari, Donatella De Zani, Davide Danilo Zani, Valeria Grieco, Chiara Giudice, Camilla Recordati, Damiano Stefanello, Luigi Auletta

**Affiliations:** 1Department of Veterinary Medicine and Animal Sciences, Università degli Studi di Milano, 26900 Lodi, Italy; 2Clinic for Small Animals Surgery, Vetsuisse Faculty, University of Zurich, CH-8057 Zurich, Switzerland

**Keywords:** veterinary oncology, sentinel lymphocentrum, sentinel lymph node, mast cell tumor, canine, intraoperative mapping, methylene blue, gamma probe

## Abstract

**Simple Summary:**

The increasing relevance given to sentinel lymph node assessment in veterinary medicine has led to the necessity of evaluating different mapping techniques and their combination. This study was performed in dogs with mast cell tumors after preoperative assessment identification of draining nodes using lymphoscintigraphy. The aim was to assess whether the addition of methylene blue dye to intraoperative detection of radioactivity in lymph nodes increased the surgical identification of such nodes. Dogs included totaled 103, for a total of 116 mast cell tumors and 196 biopsied lymph nodes. Adding the removal of blue-stained lymph nodes to radioactive positive nodes, the detection of metastatic lymph nodes increased from 90% to 95%. A low percentage of nodes detected in the single lymphocenter were both unstained and non-radioactive. Interestingly, all lymph nodes considered overtly metastatic at histopathology were radioactive. Only a few self-limiting side effects were recorded. The results of this study reported an advantage of the combination of intraoperative gamma probe and methylene blue for guiding the dissection of sentinel lymph nodes in dogs with mast cell tumors.

**Abstract:**

Methylene Blue (MB) is combined with radiopharmaceutical for intraoperative sentinel lymph node (SLN) mapping, but its role during SLN extirpation has not been investigated yet in veterinary medicine. The aim of this study was to assess whether MB increased surgical detection of SLN beyond the use of intraoperative gamma-probe (IGP) alone in clinically node-negative dogs with mast cell tumors (MCTs) following the detection of sentinel lymphocentrums (SLCs) via preoperative planar lymphoscintigraphy. Dogs enrolled underwent MCT excision and SLC exploration guided by both MB and IGP. Data recorded for each SLN were staining (blue/non-blue), radioactivity (hot/non-hot), and histopathological status (HN0-1 vs. HN2-3). A total of 103 dogs bearing 80 cutaneous, 35 subcutaneous, and 1 mucocutaneous MCTs were included; 140 SLCs were explored, for a total of 196 SLNs removed. Associating MB with IGP raised the SLNs detection rate from 90% to 95%. A total of 44% of SLNs were metastatic: 86% were blue/hot, 7% were only blue, 5% were only hot, and 2% were non-blue/non-hot. All HN3 SLNs were hot. Combining MB with IGP can increase the rate of SLN detection in dogs with MCTs; nonetheless, all lymph nodes identified during dissection should be removed, as they might be unstained but metastatic.

## 1. Introduction

Sentinel lymph node (SLN) detection and excision has become of great interest in dogs with mast cell tumors (MCTs), given the evidence of a possible lack of correspondence between the regional and sentinel lymph nodes [1,2,3,4,5] and the well-recognized role of lymph nodes assessment for staging, therapy recommendations, and prognostication [1,6,7,8,9,10]. With this aim, several mapping techniques have been described for clinical use in dogs with various malignancies, including MCT, which can be mainly categorized into preoperative and intraoperative techniques [2,3,11,12,13,14]. While preoperative mapping allows for the identification of one or more sentinel lymphocentrums (SLCs) and helps plan the surgical approach, the implementation of intraoperative guidance is beneficial as it aids the surgeon in the identification and dissection of all SLNs within the SLC. The combination of preoperative and intraoperative mapping techniques has been reported to increase the SLN detection rate, and it should be preferred to either of the methods alone [2,5,15,16].

Due to their high sensitivity and the possibility to combine pre- and intra-operative mapping phases (preoperative lymphoscintigraphy and intraoperative gamma-probing), nuclear medicine techniques based on the peritumoral administration of a radiopharmaceutical—most commonly Technetium 99 metastable (^99m^Tc)—are considered the gold standard in human medicine [17,18]. Similarly, in veterinary medicine, planar scintigraphy followed by intraoperative gamma-probing after the injection of ^99m^Tc has been proven to be highly sensitive for SLN identification [1,2,5,14]. Vital dyes, such as methylene blue (MB) and near-infrared (NIR) fluorophores, can be administered peritumorally in the same manner as radiopharmaceuticals. Since vital dyes allow for intraoperative mapping only, their use has been described alone or in combination with a preoperative technique [1,2,5,13,14,15,16,19].

The intraoperative use of the MB is well documented in canine patients, mostly as an ancillary technique used in combination with other mapping methods [1,2,15]. The main limitation to the use of MB alone, compared to NIR fluorophores, is that its signal is not visible transcutaneously before surgical incision. In addition, not all blue SLNs belonging to the same SLC can be readily recognized once surgical dissection is started due to the small size of the lymph nodes and their deep location in the adipose tissue [11,20,21]. Hence, the correct identification of the SLC with only MB can be extremely complicated if a preoperative technique is not associated [15,22,23,24] and if SLC does not correspond with the regional lymphocentrum. Thus, other intra-operative techniques, such as fluorescent lymphography with NIR fluorophores or detection with an intraoperative gamma probe (IGP), are considered superior in human literature [20,25,26]. On the other hand, the SLN detection rate of MB increases to nearly 100% when it is combined either with a preoperative or another intraoperative technique, underscoring its utility as an intraoperative guidance [1,2,5,15,19,24,27,28].

While a few studies have reported the detection rate of MB in combination with other techniques [1,2,5,15,29], there is only one available paper describing the use of MB alone in dogs and cats with spontaneous tumors [30]. However, the actual value of MB in terms of improving the IGP detection rate and the correlation between blue staining and the metastatic status of the SLN in canine patients bearing MCT has not been investigated yet.

This study aims to describe the results of the combined use of MB and IGP, after successful preoperative planar scintigraphy in dogs with MCT and non-palpable/normal-sized lymph nodes, in a 5-year, single-center experience.

## 2. Materials and Methods

Client-owned dogs referred with a cytological or histological diagnosis of MCT to the veterinary teaching hospital of the Department of Veterinary Medicine and Animal Sciences (DIVAS) of the University of Milan from January 2017 to November 2022 were included in this cross-sectional study, which included dogs enrolled in multiple prospective studies. Details of overlap are included in the acknowledgments section.

Inclusion criteria were the presence of a macroscopic MCT (cutaneous, subcutaneous, or mucocutaneous) with non-palpable and normal-sized lymph nodes without clinical or ultrasonographic evidence of suspected loco-regional metastasis; successful preoperative planar lymphoscintigraphy (at least one sentinel lymphocentrum identified); surgical excision of the primary tumor and lymphadenectomy of the SLNs guided by both IGP and MB; information about the radioactive count (RC) and presence/absence of blue staining of the SLN; histological evaluation of the excised SLN according to Weishaar et al. 2014 [6]. Exclusion criteria were presence of distant metastasis at enrollment (stage IV); T0 tumors (i.e., scars from previously excised MCT without macroscopic disease at presentation) and recurrence; regional or sentinel lymphocentrum already removed in previous surgery.

Signalment data of included cases were recorded, including breed, sex, age, and body weight. In addition, the following MCT characteristics were collected: single or multiple simultaneous presentations, anatomic location, size (greatest diameter in millimeters), and presence/absence of ulceration. Anatomical locations were categorized as limbs (below to elbow/knee); trunk (from the cranial margin of the scapula to the hip, excluding the mammary region), head and neck, inguinal, digits, mammary, and tail regions. Histological data collected from each MCT were Patnaik and Kiupel grade for cutaneous MCT [31,32], Thompson histological criteria for subcutaneous MCT [33], and surgical margins status [34,35]. Each lymph node was trimmed following the same procedure: the lymph node was divided into two halves with a longitudinal cut through the hilus. When the lymph node was thicker than 3 mm (minor axis), additional parallel cuts were performed, obtaining multiple slices (1.5 mm thick each) from each half. The whole sample was processed, and for each slice, two serial microtomic sections were cut and stained with hematoxylin–eosin and Giemsa stain, respectively. Histological evaluation of the excised SLN was performed according to Weishaar [6]. Lymph nodes and MCT grades were examined independently and blinded by three experienced pathologists (GV, GC, and RC) and revised collegially for the purpose of the study.

The SLCs were identified by preoperative planar lymphoscintigraphy and then explored with the guidance of a hand-held IGP (Crystal probe SG04; Crystal Photonic GmbH, Berlin, Germany) and MB (SALF S.p.A; Cenate Sotto, Bergamo, Italy). Briefly, patients under general anesthesia were injected with ~0.4 mL of colloidal ^99m^Tc (Nanoalbumon; Radiopharmacy Laboratory Ltd., Budaörs, Hungary) in four quadrants peritumorally. Once the SLC was identified in the planar static acquisitions, ~0.4 mL of sterile MB was injected peritumorally, as previously described [1,2,5,14]. The same injection sites were always maintained for MB and ^99m^Tc. The volume of tracers (^99m^Tc and MB) and the number of injection sites were reduced at 0,2 mL, and a single injection site for MCTs less than 0.5 cm in the maximum diameter, keeping constant the activity injected.

An SLN was considered “hot” when its RC was at least 10% of the RC recorded for the corresponding MCT. All lymph nodes identified during the dissection of the SLC (hot, blue, hot, and blue, or non-hot non-blue, but visible or palpable) were removed (Figure 1 and Figure 2). Lymphadenectomy was considered complete when the RC of the SLC was less than 10% of the hottest SLN extirped, and no other lymph nodes were visible [2,36].

The following information about the SLCs was recorded at the time of preoperative planar lymphoscintigraphy: the number of SLC for each MCT, the anatomical location, and correspondence with the regional lymphocentrum, according to Suami [37,38].

The SLCs were considered bearing metastasis if they included at least one SLN which was classified either HN2 (early metastasis) or HN3 (overt metastasis); otherwise, they were considered non-metastatic if they included only HN0 (non-metastatic) or HN1 (premetastatic) [6].

Data regarding single SLN included: total number, metastatic status, maximum diameter, anatomical location, and the correspondence with regional lymph node (as for the SLC) according to Suami et al. 2013 [37], and their radioactive and blue staining status. Blue staining (blue/non-blue) and radioactivity (hot/non-hot) of every excised SLN were evaluated ex vivo right after the extirpation [2,39], and SLNs were categorized accordingly. Their distribution within both metastatic and non-metastatic SLCs was assessed, and hence an SLC was considered:“blue” if all its SLNs were stained blue;“non-blue” if all its SLNs were not stained blue;“mixed for blue’’ if both not blue and blue SLN were present in the SLC;“hot’’ if all SLNs were hot;“non-hot’’ if all SLNs were not hot;“mixed hot’’ if both non-hot and hot SLN were detected in the same SLC.

For each SLC, the total number of lymph nodes recovered and the time to complete lymphadenectomy (from skin incision to suture for each SLC) were recorded. Finally, side effects after the injection of ^99m^TC and MB and the time of surgical excision of the MCT were reported.

### Statistical Analysis

Data recorded were imported into a dedicated software for statistical analysis (JMP^®^, v.16.0, SAS Institute, Cary, NC, USA). Descriptive statistics are reported as the number and percentage of the whole sample or specified subgroups. Normality of continuous variables was tested with the Shapiro–Wilk’s W test, and data were reported as mean ± SD or median (range) accordingly. The Mann–Whitney’s U test was applied to compare the number of SLNs for SLC between metastatic and non-metastatic SLCs, the surgical time for SLCs exploration and SLNs removal between metastatic and non-metastatic SLCs, the maximum diameter of SLNs between metastatic and non-metastatic SLNs, and the surgical time for SLC exploration and SLNs removal between non-blue–non-hot SLCs and the other categories merged. To explore whether a correlation existed between SLNs maximum diameter and bodyweight, on the overall sample and within metastatic and non-metastatic categories, a Spearman’s rank correlation coefficient (r_s_) was applied, and results were categorized according to Mukaka et al., 2012 [40].

Contingency tables, with the relative Fisher’s exact test or chi-square test, were applied to evaluate the difference in metastatic SLCs and SLNs corresponding or non-corresponding to the regional and the proportion of the blue–hot categories between metastatic and non-metastatic SLCs and SLNs. In particular, contingency tables were built up as follows: number of metastatic and non-metastatic SLCs vs. number of regional and sentinel lymphocentrums; number of metastatic and non-metastatic SLNs vs. number of regional and sentinel lymph nodes; number of metastatic and non-metastatic SLCs vs. number of lymphocentrums in each of the staining category (blue–hot, non-blue–hot, blue–non-hot, and non-blue–non-hot); number of metastatic and non-metastatic SLNs vs. number of lymph nodes in each of the staining category (blue–hot, non-blue–hot, blue–non-hot, and non-blue–non-hot); number of lymph nodes in each HN class vs. number of lymph nodes in each of the staining category (blue–hot, non-blue–hot, blue–non-hot, and non-blue–non-hot). The agreement between the two techniques (i.e., MB and IGP) was evaluated by calculating Cohen’s k statistic [41].

## 3. Results

### 3.1. Canine Patients

Data from 127 canine patients that underwent SLN biopsy were reviewed. Of those, 24 dogs were excluded for the following reasons: presentation with T0 MCT (*n* = 13) or local recurrence (*n* = 3), MB not performed (*n* = 2), missing data from IGP and MB (*n* = 2), failure to identify any SLN intraoperatively (*n* = 2), presentation with stage IV disease (*n* = 1), and owner declining the lymphadenectomy (*n* = 1). The remaining 103 canine patients were included: 24 (24%) mixed breed, 13 (13%) Labrador Retriever, 9 (8%) English Setter, 9 (8%) Golden Retriever, 7 (6%) Boxer, and 41 (40%) dogs belonging to other breeds (from 1 to 3 dogs for each breed). A total of 44 (43%) were intact males, 14 (14%) were neutered males, 37 (36%) were intact females, and 8 (8%) were spayed females. Patients were 7.4 ± 2.8 years old and had a median bodyweight of 28.1 (3–62) kg. A total of 89 dogs (86%) presented a single MCT, whereas 14 (14%) had multiple simultaneous MCTs, with a median of 2 (2–3) MCTs each. In 7 (50%) cases, the multiple MCTs were all cutaneous, in 2 (14%), all were subcutaneous, and in 5 (36%), there were simultaneously detected cutaneous and subcutaneous MCTs.

### 3.2. Mast Cell Tumors

A total of 116 MCTs were removed. The median maximum diameter was 20 (3–100) mm. Clinical and histopathological characteristics of the MCTs included are summarized in Table 1.

A single SLC drained the MCT in 91 (78%) cases; 2 SLCs were detected in 24 (21%), and 3 SLCs in 1 (1%).

### 3.3. Sentinel Lymphocentrums

A total of 140 SLCs were surgically explored, of which 49 (35%) were the superficial inguinal, 35 (25%) superficial cervical, 19 (14%) axillary, 15 (11%) popliteal, 9 (6%) accessory axillary, 6 (4%) mandibular, 5 (3%) medial iliac, 1 (1%) retropharyngeal, and 1 (1%) parotid. Overall, 62 (44%) SLCs corresponded to the expected regional lymphocentrum, whereas 78 (56%) did not. The median number of excised SLCs for each MCT was 1 (1–3). A total of 68 (49%) SLCs were metastatic, and 72 (51%) were non-metastatic. Metastatic SLCs corresponded to the regional in 31 (46%) cases and did not correspond in 37 (54%); non-metastatic SLCs corresponded to the regional lymphocentrum in 41 (57%) cases and did not correspond in 31 (43%), without any difference in the frequency of metastasis among SLCs corresponding or non-corresponding to the regional lymphocentrum (*p* = 0.76). The blue/hot categorization and their respective metastatic status are reported in Table 2.

### 3.4. Sentinel Lymph Nodes

Overall, 196 SLNs were removed from the 140 explored SLCs; 73 (37%) were classified as HN0, 37 (19%) as HN1, 66 (34%) as HN2, and 20 (10%) as HN3. Hence, 86 (44%) were metastatic (HN2-3), and 110 (56%) were non-metastatic (HN0-1). Among the 103 dogs enrolled, 54 (52%) had metastatic SLNs. The median maximum diameter of SLNs was 13 (3–49) mm; metastatic SLNs had a median diameter of 15 (4.2–45) mm, which was significantly larger (*p* = 0.04) than non-metastatic SLN (12; 3–49 mm). In the overall sample, a low correlation was detected between SLN diameter and bodyweight (r_s_ = 0.42, *p* < 0.0001); a similar trend was confirmed both for the metastatic (r_s_ = 0.44, *p* = 0.0004) and non-metastatic (r_s_ = 0.39, *p* = 0.0003) SLNs. The median number of SLNs for SLC was 1 (1–3); in metastatic SLCs, the median number of SLNs was 1 (1–3), as well as for non-metastatic SLCs (1; 1–3), with no differences in the number of SLNs between metastatic and non-metastatic SLCs (*p* = 0.10).

When considering the single SLN (Table 2), among the 86 metastatic SLNs, 37 (43%) corresponded to the regional lymph node, whereas 49 (57%) did not correspond. On the other hand, among the 110 non-metastatic SLNs, 49 (44%) corresponded to the regional lymph node, and 61 (56%) were non-corresponding, without any difference in the frequency of metastatic SLNs among corresponding or non-corresponding to the regional lymph node (*p* = 0.77). The radioactive and blue-staining SLN status divided for each histological class [6] are reported in Table 3 and Table 4. No difference was found in the distribution of the blue–hot categories between metastatic and non-metastatic SLNs (*p* = 0.24) (Table 4). Hence, metastatic SLNs were non-hot in 8 (9%) cases and non-blue in 6 (7%), whereas non-metastatic were non-hot in 12 (11%) cases and non-blue in 16 (15%), without any difference in the distribution according to the metastatic lymph nodal status (*p* = 0.68 and *p* = 0.09, respectively). The 20 HN3 SLNs were always “hot”, which differed significantly from the other histologic classes (*p* = 0.025); 18 of the 20 HN3 SLNs were also blue.

### 3.5. Surgical Data

The median surgical time for the MCT removal was 30 (10–130) minutes. The median surgical time for lymphadenectomy was 20 (5–90) minutes; for metastatic SLCs, the median surgical time was 20 (5–70) minutes, as well as for non-metastatic ones (20, 6–90 min; *p* = 0.99). The median total surgical time for the MCTs and SLNs excision was 60 (10–180) minutes. The median time of lymphadenectomy for the 3 non-blue and non-hot SLCs (25, 13–32 min) was not different compared to that elapsed to complete the lymphadenectomy in the other categories merged together (20, 5–90 min; *p* = 0.82). A total of 3 (3%) presented side effects after ^99m^Tc injection, and 11 (9%) after MB injection. In all cases, signs were mild local edema around MCT consistent with mast cell degranulation (Darier’s sign). None of the included dogs developed systemic side effects.

### 3.6. Detection Technique Agreement

Considering the SLNs, the agreement between the IGP and MB was fair (k = 0.41, *p* < 0.0001). Of the total number of lymph nodes biopsied within an SLC as identified by preoperative planar lymphoscintigraphy, the IGP alone allowed the identification of 176 (90%) “hot” SLNs, of which 78 (44%) were metastatic, and 98 (56%) were non-metastatic. With the simultaneous surgical exploration of the SLC with IGP, the MB allowed the identification of 174 (89%) “blue” SLNs, of which 80 (46%) were metastatic and 94 (54%) were non-metastatic. Therefore, their association determined the identification of 186 (95%) SLNs, of which 84 (45%) were metastatic and 102 (55%) were non-metastatic. Therefore, among the metastatic SLNs, the IGP alone would have identified 91% of SLNs within the examined SLCs, whereas the association with MB allowed the identification of 98% of metastatic SLNs.

## 4. Discussion

The feasibility and sensitivity of the various available techniques for SLN mapping is currently a matter of interest and discussion in veterinary medicine, in line with what has happened in human oncology in the past decades [19,22,42,43,44]. The impact of the SLN status on prognosis and therapeutical decision-making for several cancer types leads to the necessity to validate the best-performing mapping techniques. In the present study, the intraoperative combined use of both MB and IGP allowed for the detection of 95% of the SLNs considered sentinel based on preoperative planar lymphoscintigraphy, increasing the detection rate beyond IGP alone. Moreover, considering only the portion of metastatic SLNs detected within the examined SLCs, the addition of MB increases the detection rate to 98%. These results confirmed the value of MB in combination with IGP for the extirpation of SLN in dogs with MCT.

Methylene Blue has been proven helpful for surgeons during the learning phase of radiopharmaceutical-driven SLC dissection [45]. Nevertheless, it needs direct visualization, and it is an exclusively intraoperative technique [2,19,36], which may lead to multiple lymphocentrums or more aggressive tissue dissection to identify stained SLNs. Its association with transcutaneous mapping methods (i.e., radiopharmaceutical or NIR fluorescence with indocyanine green) helps to reduce those issues minimizing the surgical dose and the consequent risk of complications [2,5,19,27,46]. In human medicine, MB use is debated due to the possible side effects [47], which have not been reported in veterinary medicine [1,2]. Additionally, in this study, the 9% local side effect observed is likely more correlated to the degranulation following injection than MB. The reported agreement in SLN detection between MB and NIR fluorescence or lymphoscintigraphy ranges from 51 to 100% [1,2,5,24]. Lately, two articles on head and neck tumors reported the excision of a very low number of “blue” but “non-hot” or “non-fluorescent” SLNs [5,15], but all stained SLN were also reported positive for lymphoscintigraphy or NIR fluorescence in other studies [1,2,24]. In our sample, the concordance between blue staining and the radioactive count was 84%, similar to other reports in both veterinary and human literature [2,5,27,48], reaching a statistically “fair” agreement [41]. Nonetheless, the high percentage (93%) of blue SLNs reported should be carefully interpreted as we always performed a combined exploration by applying the IGP as a first-line procedure. Therefore, the detection rate of the sole use of MB could be less than what was achieved with both techniques in the present paper.

The IGP was extremely helpful during the detection of SLNs within the identified SLC by planar lymphoscintigraphy, leading to a detection rate of 91%. The measurement of the residual RC allows for straightforward decision-making during the dissection of the SLCs, as the absence of radioactivity indicates that there are no more SLNs to be removed within the SLC [36]. In fact, the absence of direct visualization of node (both blue and non-blue) does not exclude that there might be other “hot” SLNs within the same SLC [2]. The lymph nodes might be of small size, hidden within the adipose tissue, or unstained, thus making it challenging for the surgeon to achieve intraoperative identification without IGP [2] (Figure 1 and Figure 2).

Unfortunately, neither of the mapping techniques alone (MB and IGP) resulted in the “staining” of 100% of SLNs, and often the SLCs included both “blue”, “non-blue”, and “hot” and “non-hot” lymph nodes (so-called “mixed” SLCs). In the present study, 10 “non-blue” and “non-hot” lymph nodes were excised. Seven of them belonged to “mixed hot” SLCs; thus, the removal of “non-blue” and “non-hot” lymph nodes might have been achieved due to the presence of at least one adjacent hot SLN [2,5]. Indeed, sometimes only the ex vivo evaluation demonstrated that the excised node was not just the non-blue but also the “non-hot” one. In human medicine, it is reported that palpation of the SLCs for identifying further “suspicious” lymph nodes should always be accomplished, and the identified lymph nodes should be biopsied [43]. Some of the “non-hot” SLNs were early metastatic (HN2); in such cases (i.e., early metastasis), the SLNs removal has been demonstrated to have a high therapeutic impact on canine patients bearing low- and high-grade MCT [7,8,9,10].

In three separate SLCs (one mandibular and two popliteal), only one non-hot and non-blue SLN was identified. In one of these SLC (popliteal), the non-hot and non-blue SLN was bearing an early metastasis (HN2). These SLNs were detected only thanks to the preoperative planar lymphoscintigraphy that identified the SLC and to the surgical knowledge of the anatomic landmarks for lymphadenectomy. Therefore, these results confirm the importance of the combination of pre- and intra-operative mapping techniques [4,5,14,15,19,44,49]. The lack of tracers within the SLN may be due to pathophysiologic variations. In different cancer histotypes in humans, indeed, it has been suggested that lymphatic vessel embolization and the complete disruption of the SLNs anatomy might be the cause [43,44]. The hypothesis of rerouting of lymphatic flow described in human patients with extensive tumor infiltration of the draining lymph nodes [50] was not considered a matter of concern in the present study. In fact, all dogs included had normal-sized/non-palpable lymph nodes without clinical or ultrasonographic evidence of suspected loco-regional metastasis. In addition, the 20 overtly nodal metastasis reported were all occult. Moreover, considering the SLN in which there was no agreement between the IGP and MB, there might be different wash-in and wash-out times of the two tracers due to individual patients’ differences, pathophysiological phenomena, and, at least for the “non-blue–hot” category, a longer retaining time due to the colloidal formulation of the ^99m^Tc [43]. Other studies hypothesize that MB may be diluted in the lymphatic pathway and become unable to stain all the SLNs belonging to the same SLC [51]. In the study of Martin [36] on SLNs detection in human breast cancer, one-third of metastatic SLNs were not blue-stained; several possible reasons were suggested, such as subjective differences in the draining pathways linked to the distance between the injection site and the SLC, different site of injection, or even a difference in massage duration. On the other hand, for the “blue–non-hot” category, it might be hypothesized that the colloidal formulation prevents or modifies the ^99m^Tc migration, i.e., it starts migrating and tracing a direction but never reaches the SLN [36].

In the present study, SLN were categorized ex vivo for blue staining, RC, and histologically for the HN status, and the corresponding SLCs were categorized accordingly. When considering the whole sample, the rate of correspondence between the regional and the sentinel node/centrum was identical for both SLCs and SLNs. On the other hand, when considering the metastatic status, such parameters diverged between SLCs and SLNs. This is easily explained by the variable number of lymph nodes within a lymphocentrum. In the sampled population, up to three lymph nodes per SLC were removed. Although eventually the number of SLNs did not differ between metastatic and non-metastatic SLCs, the fact that more than one SLN can be found in each SLC should be kept in the surgeons’ mind during the dissection of an SLC. Moreover, multiple SLCs’ draining patterns have been described in humans [52,53], further underlying the importance of mapping techniques. Accordingly, in our sample, 21% of the MCT was drained by two SLCs and 1% by three SLCs.

Even if SLNs dimensions *per se* have been considered non-indicative of the metastatic status [2,54], in our sample, metastatic SLNs were significantly larger than non-metastatic counterparts; nonetheless, none were considered clinically enlarged. These data need further investigation, even if they are probably due to a different distribution in size of the enrolled patients among the metastatic and non-metastatic categories. The low correlation between SLN length with bodyweight in our sample, both in metastatic and non-metastatic, seems not to support the latter hypothesis, even if bodyweight might not be the most suitable variable to measure a dog’s size. Nonetheless, a correlation between bodyweight and abdominal and retropharyngeal lymph nodes has been described in dogs [55,56], and yet the latter study found an inverse correlation with age. Probably, lymph node dimensions may vary greatly in response to a large number of physio-pathological events. At least in other cancer histotypes, metastatic SLNs can be enlarged, palpable, stiffer, and more firmly anchored to the surrounding tissues [43,57]. In our sample, we could not identify any adhesion to surrounding tissues or other clinical characteristics at admission and during surgery that may have suggested the metastatic status before histopathological evaluation.

In agreement with what is reported in the literature, 54% of the enrolled dogs had early or overtly nodal metastasis [2,3,54,58]. Both the employed mapping methods (MB and IGP) confirmed the already debated discordance between the sentinel and the regional lymphocentrum [1,2,5,15,59]. Indeed, in our sample, 56% of SLCs did not correspond with the regional counterparts, underscoring once more the central role of mapping techniques in selecting the SLNs to be biopsied in dogs with MCT to avoid the risk of downstaging [1,2,3,59]. Although most of the metastatic SLC were “blue–hot” (78%), and all overtly metastatic (HN3) SLNs were “hot”, it is worth noting that 3% of all metastatic SLCs were “non-blue” and “non-hot”. Considering the therapeutic effect on HN2 and HN3 SLN removal [8,9,10], the authors propose that all SLNs encountered during the dissection of an SLC should be biopsied to avoid, again, possible downstaging as recently suggested by Alvarez-Sanchez et al. 2023 [16]. No data regarding surgical complications were reported since they completely overlap with those published by our research team [46].

This study has some limitations. First, a small number of patients were excluded to ensure a complete dataset of information. Despite this limitation, more than 100 dogs were included, and 140 SLC were explored, allowing the collection of data on almost 200 SLN. Second, we did not collect and analyze regional lymph nodes. This might have decreased the ability to exclude regional involvement. It should be carefully considered that the SLN is, by definition, the first lymph node draining a primary tumor, which harbors a higher probability of metastatic seeding. Nonetheless, it has been described that lymph nodes might be involved via the connecting tumor-associated lymphatic vasculature over non-sentinel lymph nodes within the tumor-draining basin and distant, uninvolved lymph nodes [60]. Furthermore, the presence of false-negatives for the mapping techniques described was not evaluated; however, the oncological outcome was beyond the aim of the present study, and only the lymph node staging was used.

## 5. Conclusions

The results of the present study highlight the value of MB in addition to the IGP during intraoperative detection of SLNs within SLCs, previously detected by preoperative planar scintigraphy. Its value is emphasized by the increased detection rate, the absence of side effects, and the low cost of MB. Based on the finding of a small percentage of metastatic non-blue and non-hot SLNs, neither the absence of MB staining nor of ^99m^Tc during SLN removal guarantees their metastatic status. Hence, all SLNs identified in an SLC should be removed and histologically evaluated.

## Figures and Tables

**Figure 1 animals-13-01854-f001:**
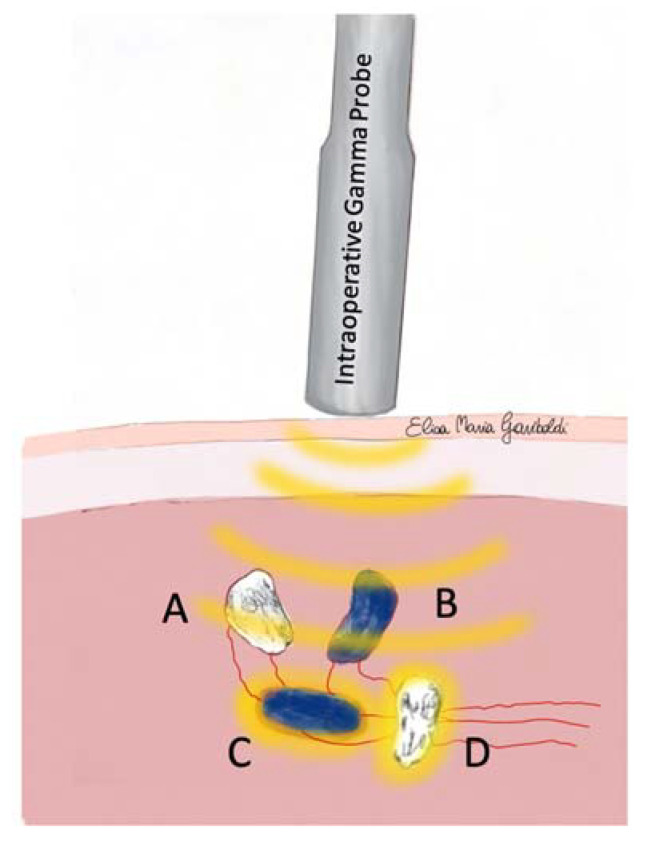
Intraoperative scenario of SLC exploration using IGP with at least one “hot and blue” (C) or “hot and non-blue” (D) SLN deeper than “non-hot” SLNs (A and B). The surgeon removes all lymph nodes encountered during sentinel lymphocentrum exploration (both “non-hot” and “blue”, or “non-hot” and “non-blue”) until RC will be less than 10% of the hottest SLN extirped and no other lymph nodes are visible/palpable.

**Figure 2 animals-13-01854-f002:**
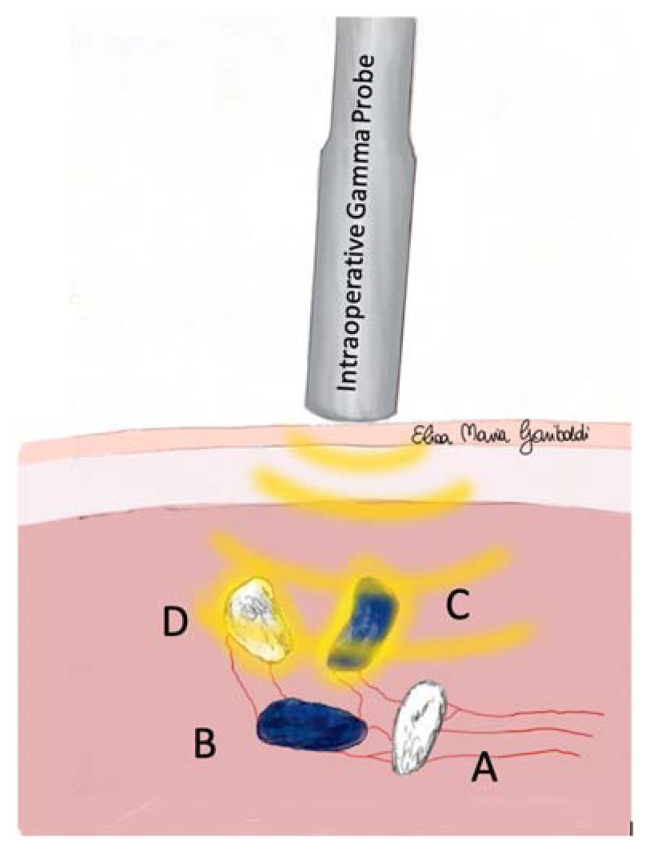
Intraoperative scenario of SLC exploration using IGP, with at least one “hot and blue” (C) or “hot and non-blue” (D) SLN less deep than other non-hot SLNs (A and B). The surgeon removes the “hot” lymph nodes encountered during sentinel lymphocentrum exploration and all other lymph nodes identified using palpation or because they are very close (A) or visible (B) to hot nodes (C,D). The surgeon evaluates the field RC at the end of the surgical exploration. The exploration is stopped when the RC is less than 10% of the hottest SLN extirped and no other lymph nodes are visible/palpable.

**Table 1 animals-13-01854-t001:** Clinical and histopathological characteristics of MCT.

Variables	N° of Tumors and %
**Clinical presentation**	
Anatomical localization	
Limbs	37 (32%)
Trunk	33 (28%)
Head and neck	16 (14%)
Inguinal region	15 (13%)
Digits	8 (7%)
Mammary region	5 (4%)
Tail	2 (2%)
Ulceration	
Yes	18 (16%)
No	98 (84%)
**Histopathological data**	
Histopathological type	
** Cutaneous**	**80 (69%)**
Patnaik	
I	15 (19%)
II	64 (80%)
III	1 (1%)
Kiupel	
Low-grade	78 (97%)
High-grade	2 (3%)
** Subcutaneous**	**35 (30%)**
Histological pattern	
Infiltrative	19 (54%)
Circumscribed	4 (12%)
Combined	12 (34%)
** Mucocutaneous**	**1 (1%)**
Surgical margins	
Clean	90 (78%)
Clean but close	3 (2%)
Infiltrated	23 (20%)

**Table 2 animals-13-01854-t002:** Number (percentage) of sentinel lymphocentrums belonging to the blue–hot categories, divided by metastatic status.

	Blue–Hot	Blue–Non-Hot	Blue –Mixed Hot	Non-Blue –Hot	Non-Blue –Non-Hot	Non-Blue–Mixed Hot	Mixed Blue –Hot	Mixed Blue–Non-Hot	Mixed Blue–Mixed Hot	Total
Non Metastatic	60 (83%)	1 (1.5%)	0	5 (7%)	2 (3%)	1 (1.5%)	0	0	3 (4%)	72 (100%)
Metastatic	53 (78%)	3 (4%)	5 (7.5%)	2 (3%)	1 (1.5%)	1 (1.5%)	1 (1.5%)	0	2 (3%)	68 (100%)
Total	113 (80.7%)	4 (2.9%)	5 (3.6%)	7 (5%)	3 (2%)	2 (1.4%)	1 (0.7%)	0	5 (3.6%)	140 (100%)

**Table 3 animals-13-01854-t003:** Distribution of SLN for each histological nodal class [6] and radioactive and staining status.

	Blue–Hot	Blue–Non-Hot	Non-Blue–Hot	Non-Blue–Non-Hot	Total
**HN0**	56 (76.7%)	4 (5.5%)	6 (8.2%)	7 (9.6%)	73 (100%)
**HN1**	34 (91.9%)	0	2 (5.4%)	1 (2.7%)	37 (100%)
**HN2**	56 (84.9%)	6 (9.1%)	2 (3%)	2 (3%)	66 (100%)
**HN3**	18 (90%)	0	2 (10%)	0	20 (100%)
**Total**	164 (84%)	10 (5%)	12 (6%)	10 (5%)	196 (100%)

**Table 4 animals-13-01854-t004:** Distribution of SLN categorized into non-metastatic (HN0-HN1) and metastatic (HN2-HN3) and radioactive and staining status.

	Blue–Hot	Blue–Non-Hot	Non-Blue–Hot	Non-Blue–Non-Hot	Total
**HN0-HN1** **(non-metastatic)**	90 (82%)	4 (4%)	8 (7%)	8 (7%)	110 (100%)
**HN2-HN3** **(metastatic)**	74 (86%)	6 (7%)	4 (5%)	2 (2%)	86 (100%)
**Total**	164 (84%)	10 (5%)	12 (6%)	10 (5%)	196 (100%)

## Data Availability

The data that support the findings of this study are available from the corresponding author upon reasonable request.

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
