# Peer review of "Evaluation of Surgical Aid of Methylene Blue in Addition to Intraoperative Gamma Probe for Sentinel Lymph Node Extirpation in 116 Canine Mast Cell Tumors (2017–2022)"

_animals, 2023, doi:10.3390/ani13111854_

Round 1

Reviewer 1 Report

The manuscript presents a retrospective study of usefulness of methylene blue in addition to intraoperative gamma probe for sentinel lymph nodes extirpation in dogs suffering from mast cell tumors. The manuscript is interesting and can usefully contribute to the diagnosis and surgical procedures of tumor treatment. The material and methods as well as the results are described in detail. However, in order to make the text easier to understand and follow, I would definitely recommend displaying the results with a few more tables or possibly a graph. Namely, when the results are presented mostly only in text, it is difficult to follow and understand them well. Only two tables are presented, with the fact that in Table 2 it should be indicated that the results are presented in percentages.

Author Response

The Authors are grateful to Reviewers for their questions and suggestions that helped improving the paper. The Authors thank the Reviewers for the time dedicated to their paper.

REV1

Comments and Suggestions for Authors

The manuscript presents a retrospective study of usefulness of methylene blue in addition to intraoperative gamma probe for sentinel lymph nodes extirpation in dogs suffering from mast cell tumors. The manuscript is interesting and can usefully contribute to the diagnosis and surgical procedures of tumor treatment. The material and methods as well as the results are described in detail. However, in order to make the text easier to understand and follow, I would definitely recommend displaying the results with a few more tables or possibly a graph. Namely, when the results are presented mostly only in text, it is difficult to follow and understand them well. Only two tables are presented, with the fact that in Table 2 it should be indicated that the results are presented in percentages.

Authors: The Authors thank the Reviewer for the suggestion. We added a table to summarize MCT characteristics (clinical and histopathological). Table 2 was not expressed in percentage. To make Table 2 (now Table 3) easier to understand, Authors have now added the percentage divided by HN. Next, the Authors added Table 4, dividing HN classes into metastatic and non-metastatic. The Authors subsequently slightly modified the relative text.

Reviewer 2 Report

This is a retrospective study detailing the use of concurrent lymphoscintigraphy (both pre-operative and intra-operative) and methylene blue in 116 dogs. The study reports a high rate of identification of a sentinel node using both techniques, with strong but incomplete agreement between techniques. 

The main limitation of this study, in isolation, is the lack of any assessment beyond the identified sentinel lymph node(s). This must necessarily limit the conclusions that can be drawn as there are no data regarding false negative rates of these techniques. The disagreement between techniques shown in multiple studies, including this one, suggests an impact of technique, and yet the authors seem to imply (as many sentinel lymph node studies do) that all results should be taken at face value and that any node identified is valid. As the authors note, multiple factors, including regional tumor progression, but perhaps even aspects of technique, such as positioning, could impact the node or nodes identified. This work currently does not adequately acknowledge these limitations. A relatively high proportion of identified nodes were considered not to represent the regional anatomic center or node; however, the methodology for determination of this in a retrospective study is insufficiently clear. In addition, it is possible that the expected node based on drainage patterns may be more reflective of regional metastasis in some dogs (such as if extensive nodal metastasis results in obstructing and rerouting of lymphatic flow), but this possibility is largely ignored, aside from a brief comment in the latter part of the discussion. Given these concerns, a particularly important consideration is whether dogs with clinical evidence of nodal metastasis were included, or if this study was restricted solely to dogs without evidence of nodal metastasis pre-operatively. If dogs with suspected nodal metastasis were included, details about the pre-operative assessment of these cases must be included.

The authors are clearly highly active in the field of sentinel lymph node techniques and have contributed substantially to the body of evidence regarding some of these techniques in veterinary medicine. This study, however, appears to have substantial overlap with previous publications from the same group, including the following: 

Ferrari et al. Biopsy of sentinel lymph nodes after injection of methylene blue and lymphoscintigraphic guidance in 30 dogs with mast cell tumors. Vet Surg. 2020

Ferrari et al. Assessing the Risk of Nodal Metastases in Canine Integumentary Mast Cell Tumors: Is Sentinel Lymph Node Biopsy Always Necessary? Animals. 2021

Manfredi et al. Preoperative planar lymphoscintigraphy allows for sentinel lymph node detection in 51 dogs improving staging accuracy: Feasibility and pitfalls. Vet Radiol Ultrasound. 2021

Chiti et al. Surgical complications following sentinel lymph node biopsy guided by γ-probe and methylene blue in 113 tumour-bearing dogs. Vet Comp Oncol. 2023

Given the relatively short time period for case accrual, overlap in published study enrolment periods, overlap in authors and details, it seems likely that this retrospective study actually includes a large number of dogs enrolled in previously published prospective studies. Was this the case? If so, this should at least be clearly acknowledged, including details of how many dogs were included from each publication. I also feel that the authors need to more clearly justify the rationale for publishing this cohort. While this is a relatively large population for veterinary medicine managed with consistent sentinel lymph node techniques, it lacks a control and the utility of additional cases to build an evidence base decreases if these are actually often the same as those published. Duplicate publication risks a limited population of dogs being interpreted as part of a more extensive evidence base, implying greater support of conclusions than ideal, particularly with repeated self-citation. The exaggeration of effects could be particularly problematic during attempts at systematic review. 

The standard of writing is fair but I feel this manuscript would benefit from review and editing by a native English speaker. While a few areas of language are mentioned below, this is not exhaustive, as my focus should be primarily scientific content.  

Specific Comments:

Lines 2-4: I’m not convinced you can describe this study as assessing the “usefulness” of methylene blue in addition to lymphoscintigraphy without a control, active long-term follow-up to reliably identify sites of regional failure, or effects on outcome. 

Lines 19-20: It is unclear from this summary how a node could be sentinel and yet both unstained and non-radioactive.

Lines 141-143: “The SLCs…” to “…metastasis) SLN [6];” – Review phrasing.

Line 146: How was the regional lymph node determined retrospectively? Please provide more detail.

Lines 177-180: Please provide a more detailed breakdown of this component of the statistical analysis.

Line 188: This is a retrospective study, so these dogs weren’t really enrolled, at least as currently described. 

Lines 193-195: A list of every breed included seems excessive. Just the most common breeds should suffice. 

Line 196: Please specify intact male and intact female for clarity when reading.

Lines 203-204: How were distinctions made for borderline lesions (e.g. limb v. digit or limb v. inguinal)?

Lines 219-220: Did you do anything to investigate the 78 lesions where the identified lymph center didn’t correspond to the regional lymph center? It seems like not but, if that is the case, this omission critically weakens the study.

Lines 271-272: The patients cannot report systemic side effects themselves. Please reconsider phrasing. 

Lines 288-290: “In the present…” to “IGP alone.” If my understanding is correct, this statement should be amended to indicate that these percentages are for nodes identified as sentinel based on pre-operative lymphoscintigraphy. As there is no additional assessment or control, the blanket statement currently used is inappropriate.

Lines 290-291: The authors state a detection rate of metastatic LNs with methylene blue of 98% but there is no valid control or additional steps to support this percentage as currently phrased. The current wording of this sentence and the former markedly misrepresent the work as performed. I don’t believe that this is intentionally misleading, but more indicative of the authors’ enthusiasm for the techniques; however, there is a profound difference between finding that 98% of the metastatic nodes you happened to look at using some related techniques were identified and being able to confidently state that 98% of all metastatic nodes were identified. 

Lines 303-305: If data are not provided back up this statistic, please do not state a specific figure.

Lines 293-333: In a laudable effort to be thorough, the authors’ message has become more challenging for the reader to discern. I would suggest reviewing this paragraph and editing it to become more concise and focused.

Lines 328-333: The use of methylene blue in Rossanese’s study is distinctly different to the use of methylene blue in this study. It is inappropriate to try to draw comparisons in rates of success here.

Lines 334-335: Again, it should be specified whether this is 91% of dogs that had a node identified, or whether it allowed identification of 91% of nodes identified on pre-operative lymphoscintigraphy. Regardless, neither is the same as a blanket sensitivity percentage, as that would require additional exploration of regional nodes. 

Lines 355-357: Please review this first sentence for phrasing.

Lines 361-363: Absolutely, and yet this possibility is largely sidelined for most of the paper.

Lines 399-404: A similar time for resection cannot be used as an indication that nodes were not fixed. It may well have been that no nodes were fixed but the time of resection is not an appropriate marker of this.

Lines 419-423: The limitations section is insufficient based on the study design and should more thoroughly explore the limitations of a retrospective study, without any form of control or assessment for false negatives. 

Author Response

The Authors are grateful to Reviewers for their questions and suggestions that helped improving the paper. The Authors thank the Reviewers for the time dedicated to their paper.

REV2

This is a retrospective study detailing the use of concurrent lymphoscintigraphy (both pre-operative and intra-operative) and methylene blue in 116 dogs. The study reports a high rate of identification of a sentinel node using both techniques, with strong but incomplete agreement between techniques. 

The main limitation of this study, in isolation, is the lack of any assessment beyond the identified sentinel lymph node(s). This must necessarily limit the conclusions that can be drawn as there are no data regarding false negative rates of these techniques. The disagreement between techniques shown in multiple studies, including this one, suggests an impact of technique, and yet the authors seem to imply (as many sentinel lymph node studies do) that all results should be taken at face value and that any node identified is valid. As the authors note, multiple factors, including regional tumor progression, but perhaps even aspects of technique, such as positioning, could impact the node or nodes identified. This work currently does not adequately acknowledge these limitations. A relatively high proportion of identified nodes were considered not to represent the regional anatomic center or node; however, the methodology for determination of this in a retrospective study is insufficiently clear. In addition, it is possible that the expected node based on drainage patterns may be more reflective of regional metastasis in some dogs (such as if extensive nodal metastasis results in obstructing and rerouting of lymphatic flow), but this possibility is largely ignored, aside from a brief comment in the latter part of the discussion.

.

Authors: The aim of this study was not to find out whether methylene blue is superior to scintigraphy (in both preoperatory and intraoperative phases) in identifying a certain number of SLNs: indeed, the study describes the results of the combined use of MB and IGP, after preoperative planar scintigraphy, during SLNs detection in dogs with MCT. The study was not designed to explore the impact on the clinical outcome of this technique. Furthermore, our study did not aim to make a comparison on the detection performance of SLN between methylene blue and scintigraphy, but rather to evaluate, for all single cases in which preoperative scintigraphy revealed the lymphocentrum, how many sentinel lymph node/s staining blue dye and radioactive were detected and how these characteristics correlated to the presence of lymph node metastasis according to Weishaar classification. The Authors agree with REV2 that no studies compare intraoperative and intra-patient techniques. It is necessary to understand which technique has a lower false negative rate and evaluate the impact on clinical outcome, and that could be done using target events like nodal relapse, distant relapse, etc., during a follow-up period. However, such a comparison of the two techniques should be done in the preoperative and intraoperative phases. The blue dye alone cannot be considered a preoperative mapping and intraoperative guidance because it is only visible after dissection.

Furthermore, blue staining alone as an intraoperative guide is not enough to guide the surgeon in the decision making on when to interrupt the surgical exploration, i.e. when all the draining lymph node have been excised or if he/she has to look for further nodes. For these reasons and based on the declared aims, this study cannot provide any data on the comparison between techniques’ performances. The comments provided by REV2 should certainly be explored and pursued in a different new study. The Authors did not remove the regional lymph node if there was no agreement between regional and sentinel lymph nodes. As shown in previous experimental studies in dogs, this technique has been shown to identify correctly SLN (Aquino et al. Acta Cir Bras. 2012;27(2):102-8. doi: 10.1590/s0102; Suami et al. PLoS One. 2013;8(7):1-9). In addition, a recent study reported that removing the regional lymph node and the sentinel lymph node is unnecessary to exclude a possible false negative (Ferraris et al. Vet Comp Oncol. 2023 Jan 12. doi: 10.1111/vco.12878). The Authors added the latter reference in the discussion section (lines 632-635).

In this study, the Authors removed all detected lymph nodes allocated in the lymphocentrum identified by the combined use of preoperative lymphoscintigraphy and intraoperative gamma probe plus blue dye because, in the presence of more than one lymph node, these may present different Weishaar metastatic status.  Consequently, to reduce the risk of missing nodal metastasis, all SLNs were extirpated as previously suggested (Ferrari et al. Vet Surgery 2020; Alvarez-Sanchez A, et al. Vet Surg. 2023). The results of this study (based on a large sample population) are consistent with that previous study. Lastly, our study reported that 10 out of 196 SLNs extirpated were non-hot and non-blue, 2 of which had early nodal metastasis (this information is present in the submitted paper)

We did not discuss positioning because, according to the inclusion criteria, in all cases enrolled, preoperative mapping with planar scintigraphy and gamma probe must have identified at least a lymphocentrum. We specified at line 91-95 (aim) and 106-108 (inclusion criteria) that only dogs with successful preoperative lymphoscintigraphy were included. In addition, positioning has not been yet reported influencing the successful of lymphoscintigraphic and methylene blue procedures as, instead, was reported for NIRF-ICG (Beer et al, ECVS congress). 

Considering that the study includes dogs with no palpable and normal-sized lymph nodes without clinical or ultrasonographic evidence of suspected locoregional metastasis, the hypothesis that extensive nodal metastasis could have modified the lymphatic flow was not stressed in the discussion because in all cases included preoperative lymphoscintigraphy identified at least one lymphocentrum and occult overtly nodal metastasis were diagnosed in 20/196 SLN. We add a new sentence at lines 571-576: the hypothesis of rerouting lymphatic flow was not considered a possible explanation because the Authors included all dogs with no palpable, normal-size lymph nodes without clinical or ultrasonographic evidence or suspected regional metastasis. Also, in all cases included, preoperative lymphoscintigraphy identified the lymphocentrum, and occult overtly nodal metastasis was diagnosed in 20 out of 196 SLN (Goyal, Douglas-Jones AG, Newcombe RG, Mansel RE. Effect of lymphatic tumor burden on sentinel lymph node biopsy in breast cancer. Breast J. 2005 May-Jun;11(3):188-94. doi: 10.1111/j.1075-122X.2005.21591.x.

Given these concerns, a particularly important consideration is whether dogs with clinical evidence of nodal metastasis were included, or if this study was restricted solely to dogs without evidence of nodal metastasis pre-operatively. If dogs with suspected nodal metastasis were included, details about the pre-operative assessment of these cases must be included.

Authors: All dogs included in the study have no palpable and/or normal size lymph nodes without clinical or ultrasonographic evidence or suspected regional metastasis. The Authors apologize for this oversight which made the aim and discussion unclear. This information was missing in the submitted manuscript, and was added at lines 92-93 and 106-107 .To better clarify to Reviewer 2, the Authors would like to underline that in the presence of no palpable or normal size lymph nodes without clinical or ultrasonographic evidence or suspected regional metastasis, systematic cytological exclusion of regional lymph nodes metastasis was not performed for the following reasons: in the presence of lack of correspondence between regional and sentinel lymph node without mapping, in presence of multiple lymph nodes belonging to the lymphocentrum the true SLN could not be sampled or the cytological status of one and more accessible lymph node could be not representative of all the nodes, and, as previously reported, patterns of imaging features obtained by fine needle aspiration are insufficient to determine with high repeatability the status of the lymphatic basin in dogs with MCT  (Ku et al., Vet. Comp. Oncol. 2017, 15, 1206–1217, doi:10.1111vco.12256; Fournier et al Vet. Clin. Pathol. 2018, 47, 489–500, doi:10.1111/vcp.12636; Fournier et al. Vet. Comp. Oncol. 2020, doi:10.1111/vco.12647).

The authors are clearly highly active in the field of sentinel lymph node techniques and have contributed substantially to the body of evidence regarding some of these techniques in veterinary medicine. This study, however, appears to have substantial overlap with previous publications from the same group, including the following: 

Ferrari et al. Biopsy of sentinel lymph nodes after injection of methylene blue and lymphoscintigraphic guidance in 30 dogs with mast cell tumors. Vet Surg. 2020

Ferrari et al. Assessing the Risk of Nodal Metastases in Canine Integumentary Mast Cell Tumors: Is Sentinel Lymph Node Biopsy Always Necessary? Animals. 2021

Manfredi et al. Preoperative planar lymphoscintigraphy allows for sentinel lymph node detection in 51 dogs improving staging accuracy: Feasibility and pitfalls. Vet Radiol Ultrasound. 2021

Chiti et al. Surgical complications following sentinel lymph node biopsy guided by γ-probe and methylene blue in 113 tumour-bearing dogs. Vet Comp Oncol. 2023

Given the relatively short time period for case accrual, overlap in published study enrolment periods, overlap in authors and details, it seems likely that this retrospective study actually includes a large number of dogs enrolled in previously published prospective studies. Was this the case? If so, this should at least be clearly acknowledged, including details of how many dogs were included from each publication. I also feel that the authors need to more clearly justify the rationale for publishing this cohort. While this is a relatively large population for veterinary medicine managed with consistent sentinel lymph node techniques, it lacks a control and the utility of additional cases to build an evidence base decreases if these are actually often the same as those published. Duplicate publication risks a limited population of dogs being interpreted as part of a more extensive evidence base, implying greater support of conclusions than ideal, particularly with repeated self-citation. The exaggeration of effects could be particularly problematic during attempts at systematic review. 

Authors: The mapping and extirpation of SLN is a recent field of research in canine and feline surgical oncology, and scientific information is scarce and often based on a small sample population.

The research project on SLNs started in the academic institution of the Authors in 2017 thanks to strict cooperation between researchers working in multidisciplinary and interdisciplinary areas of veterinary oncology such as surgery, pathology and radiology. All these researchers contributed significantly to the progressive enrolment of clinical cases, permitting us to count on solid and repetitive inclusion and exclusion criteria that have enabled us to collect a homogenous and large sample population following the most recent publication and literature (self-citations excluded).

This approach allowed our research group to investigate and reply to some questions born in the last five years within the international veterinary community dedicated to canine oncology, with publications through the years in several peer-reviewed international Journals and with different aims for each scientific question.

As required by REV2, the Authors reported the distribution of cases in previous papers. However, the Authors believe it is unnecessary to declare in this paper the overlap of some of the cases because previous studies had different aims. In addition, the Authors already underlined that in the discussion, where the overlapping of cases of this manuscript was reported with Chiti et al. 2023, who, however, has a different aim (surgical complications in SLN).

Present paper: 103 dogs and 116 MCT

Ferrari et al. 2020 Vet surgery: 30 dogs and 34 MCT

Ferrari et al. 2021 Animals: 53 dogs 66 MCT

Manfredi et al 2021 Vet Radiology and Ultrasound:  47 dogs with 47 MCT in a case sample of 60 solid tumors

Chiti et al. 2020 Vet Comp Oncology:  98 out of 113 cases were MCT

The Authors think that talking about "duplicate publication" is unfair for this research group since each study previously published has a specific aim applied in a specific population with rigid and repetitive inclusion and exclusion criteria supported by previous publications, most of which were produced by researchers outside at this research group. Furthermore, even if they come from the same institution but individually address specific aims, data published in peer-reviewed scientific journals have a higher scientific value than personal opinions about SLNs presented as invited speakers at international congresses or published reviews.

When considering self-citation, the Authors agree with the Reviewer. Some consideration about the Reviewer's concerns is needed: the lymphoscintigraphy technique (preoperative and intraoperative phases) requires specific and expensive facilities and equipment. The authors do not have a precise idea of how many veterinary facilities (academic and non-academic) in the world have this technology, but it is safe to assume that there are only a few, perhaps no more than ten, worldwide. The reason why other institutions, which are undoubtedly stronger in clinical research in veterinary oncology and that can count on a much larger number of cases than ours, do not publish the results of their research on the same technique is unknown to us, and this is a variable that the Authors cannot control. Some of these institutions have published in past numerically smaller case series of research on SLNs, but ten years later, have not produced any new papers yet. The paucity of publications on the use of lymphoscintigraphy in the SLN led us to cite ourselves. However, the Authors think that all self-citation is correct, and so it is for our conduct.

The Reviewer claims that this study lacks control, but, as mentioned before, our study did not compare detection rate of SLN between methylene blue and scintigraphy. It aimed to evaluate in the individual case (when the preoperative scintigraphy revealed the lymphocentrum) how many sentinel lymph node/s staining blue dye and radioactive were detected and how this correlated with the presence of lymph node metastasis according to the Weishaar classification. Comparing the two techniques requires that they are comparable both preoperatively and intraoperatively. The blue dye alone cannot be considered enough for preoperative mapping because it is only visible after dissection. Further, methylene blue alone is not a proper intraoperative guide since it does not help the surgeon in finding the landmarks for the dissection of the lymphocentrum (i.e., it cannot be seen transcutaneously before skin incision) and, in addition, it does not help to decide when to interrupt the surgical exploration if the next SLN are not directly visible. For these reasons and based on the declared aims, this study cannot provide any data on the compared performances  of the techniques. Also, if there was no agreement between the regional and sentinel lymph nodes, the Authors did not remove the regional lymph node. As shown in previous experimental studies on dogs, these techniques can identify correctly SLN (Aquino et al. Acta Cir Bras. 2012;27(2):102-8. doi: 10.1590/s0102; Suami H et al. PLoS One. 2013;8(7):1-9). In addition, a recent study shows that removing the regional lymph node and the sentinel lymph node is unnecessary to exclude a possible false negative. (Ferraris E et al. Vet Comp Oncol. 2023 Jan 12. doi: 10.1111/vco.12878).

The standard of writing is fair but I feel this manuscript would benefit from review and editing by a native English speaker. While a few areas of language are mentioned below, this is not exhaustive, as my focus should be primarily scientific content. 

Authors: the Authors answered and corrected all specific comments and revised English

Specific Comments:

Lines 2-4: I’m not convinced you can describe this study as assessing the “usefulness” of methylene blue in addition to lymphoscintigraphy without a control, active long-term follow-up to reliably identify sites of regional failure, or effects on outcome. 

Authors: The study intended to assess how MB added to IGP may increase detection rate of SLN, associated also to the staging value of this increased number of SLN extirpated. All lymph node with a radioactive count and/or blue-stained and/or directly visible in the lymphocentrum detected by pre-operative planar scintigraphy were removed. The control is represented by the number of SLN “hot” that do not also need the blue-staining to be detected. The clinical outcome of patients is beyond the aim of this study; please see the previous author's reply.  

Lines 19-20: It is unclear from this summary how a node could be sentinel and yet both unstained and non-radioactive. 

Authors: The lymph node was defined as sentinel because preoperative planar scintigraphy identified the surgically explored SLC, as reported in Material and Method. More details are presented in the text because the simple summary allows a limited number of words (guidelines of the journal)

The sentence was changed: “A low percentage of nodes detected in the single lymphocenter resulted in both unstained and non-radioactive.” (lines 19-20)

Lines 141-143 (now 182-185): “The SLCs…” to “…metastasis) SLN [6];” – Review phrasing.

Authors: “The SLCs were considered metastatic if including at least one HN2 (early metastasis) or HN3 (overt metastasis) SLN [6]” was changed: “The SLCs were considered bearing metastasis if they included at least one SLN which was classified either HN2 (early metastasis) or HN3 (overt metastasis) [6];”

Line 146: How was the regional lymph node determined retrospectively? Please provide more detail.

Authors: Regional lymph nodes were defined according to Suami et al. (ref. 37) as reported in the submitted manuscript at line 181. RLN was not determined retrospectively but it was reported in the clinical record at time of case admission based of Suami et al, 2013. The correspondence between SLC and RLN was evaluated during the preoperative phase (planar lymphoscintigraphy) and intraoperative phase (gamma probe) in each of the 116 cases, as reported in our previous published papers.

The sentence was modified as follows “Data regarding single SLN included: total number, metastatic status, maximum diameter, the anatomical location and the correspondence with regional lymph node according to Suami et al, 2013. (now lines 186-188)

Lines 177-180: Please provide a more detailed breakdown of this component of the statistical analysis.

Authors: The Authors thank the Reviewer for the suggestion, and we explained how the contingency tables were built up for readers’ convenience.

We add in the text this new sentence (Lines 221-230)  “In particular, contingency tables were built up as follows: the number of metastatic and non-metastatic SLCs vs. the number of regional and sentinel lymphocentrums; the number of metastatic and non-metastatic SLNs vs. the number of regional and sentinel lymph nodes; the number of metastatic and non-metastatic SLCs vs. the number of lymphocentrums in each of the staining category (blue–hot, non-blue–hot, blue–non-hot, non-blue–non-hot); the number of metastatic and non-metastatic SLNs vs. the number of lymph nodes in each of the staining category (blue–hot, non-blue–hot, blue–non-hot, non-blue–non-hot); the number of lymph nodes in each HN class vs. the number of lymph nodes in each of the staining category (blue–hot, non-blue–hot, blue–non-hot, non-blue–non-hot).”

Line 188 (now 251): This is a retrospective study, so these dogs weren’t really enrolled, at least as currently described. 

Authors: changed it in the text with: “included”

Lines 193-195 (now 251-254): A list of every breed included seems excessive. Just the most common breeds should suffice. 

Authors: Done. “The remaining 103 canine patients were included: 24 (24%) mixed breed, 13 (13%) Labrador Retriever, 9 (8%) English Setter, 9 (8%) Golden Retriever, 7 (6%) Boxer, and 41 (40%) dogs belonging to other breeds (from 1 to 3 dogs for each breed).”

Line 196 (now 254-255): Please specify intact male and intact female for clarity when reading.

Authors: We added this modified sentence “Forty-four (43%) were intact males, 14 (14%) were castrated males, 37 (36%) were intact females, and 8 (8%) were spayed females”

Lines 203-204: How were distinctions made for borderline lesions (e.g. limb v. digit or limb v. inguinal)?

Authors: The Authors used the same anatomical classification widely used in several other papers focused on canine MCT, for example Fejos et al. Vet comp oncology 2021. Specified it in the text in material and methods (lines 118-120) Anatomical localizations included: limbs (below to elbow/knee) 37 (32%), trunk (from cranial margin of the scapula to the hip, excluded the mammary region) 33 (28%), head and neck region 16 (14%), inguinal region 15 (13%), digits 8 (7%), mammary region 5 (4%), tail 2 (2%).

Lines 219-220: Did you do anything to investigate the 78 lesions where the identified lymph center didn’t correspond to the regional lymph center? It seems like not but, if that is the case, this omission critically weakens the study.

Author: In case there was no agreement between regional and sentinel lymph nodes, the Authors did not remove the regional lymph node. As shown in previous experimental studies on dogs, these techniques can identify correctly SLN (Aquino et al. Acta Cir Bras. 2012;27(2):102-8. doi: 10.1590/s0102; Suami H et al. PLoS One. 2013;8(7):1-9). In addition, a recent study shows that removing the regional lymph node and the sentinel lymph node is unnecessary to exclude a possible false negative. (Ferraris E et al. Vet Comp Oncol. 2023 Jan 12. doi: 10.1111/vco.12878).

Lines 271-272: The patients cannot report systemic side effects themselves. Please reconsider phrasing. 

Author: Done. Changed it in the text as follows: “none of the included dogs developed systemic side effects.”

Lines 288-290: “In the present…” to “IGP alone.” If my understanding is correct, this statement should be amended to indicate that these percentages are for nodes identified as sentinel based on pre-operative lymphoscintigraphy. As there is no additional assessment or control, the blanket statement currently used is inappropriate.

Author: Lymphoscintigraphy is a widely used technique, especially in women's breast cancer, because of its dual value as both a pre-and intra-operative technique, as also recently reported in a publication of our research group (Manfredi et al. 2021). The preoperative planar lymphoscintigraphy identifies the SLC but it’s not able to detect the number of SLN included in that SLC. Number of SLN that were detected by the IGP, instead. We did not specify this information because it is widely known in human medicine and reported in previous veterinary literature. The present study did not compare performance to detect SLN between methylene blue and scintigraphy (including pre- and intra-operative phases). However, we evaluated in each case in which the preoperative scintigraphy revealed the lymphocentrum and the dog was admitted to SLN biopsy, how many radioactive and/or staining blue sentinel lymph node/s were detected to understand the usefulness of the addition of the blue-dye, and how these correlated to the presence of lymph node metastasis according to Weishaar grades.

Lines 290-291: The authors state a detection rate of metastatic LNs with methylene blue of 98% but there is no valid control or additional steps to support this percentage as currently phrased. The current wording of this sentence and the former markedly misrepresent the work as performed. I don’t believe that this is intentionally misleading, but more indicative of the authors’ enthusiasm for the techniques; however, there is a profound difference between finding that 98% of the metastatic nodes you happened to look at using some related techniques were identified and being able to confidently state that 98% of all metastatic nodes were identified. 

Author: In this study, the Authors removed all lymph nodes allocated in the lymphocentrum identified by the combined use of preoperative lymphoscintigraphy that were “hot” using the intraoperative gamma probe identification and/or blue-stained and/or directly visible in the surgical field. In the presence of more than one lymph node, these may show a different Weishaar grade. Consequently, to reduce the risk of missing nodal metastasis, all nodes in the SLCs were extirpated as previously suggested (Ferrari et al. Vet Surgery 2020; Alvarez-Sanchez et al. Vet Surg. 2023). Results of the present study based on a large sample population are consisted with previous studies. Further, our study reported that 10 out of 196 SLNs extirpated were non-hot and non-blue, 2 with early nodal metastasis (this information is present in the submitted paper).

The percentage reported was related to all SLNs biopsied in the SLC identified with preoperative lymphoscintigraphy and surgical explored with gamma probe plus MB.

We slightly modified the text to clarify how the study assessed the combined use of MB and IGP. “In the present study, the intraoperative combined use of both MB and IGP allowed for the detection of 95% of the SLNs, increasing the detection rate of IGP alone. Moreover, considering only the portion of metastatic SLNs detected, the addition of MB permits to increase the detection rate to 98%.” (lines 413-417)

Lines 303-305: If data are not provided back up this statistic, please do not state a specific figure.

Authors: The sentence was removed.

Lines 293-333 (now 419-439): In a laudable effort to be thorough, the authors’ message has become more challenging for the reader to discern. I would suggest reviewing this paragraph and editing it to become more concise and focused.

Author: the paragraph was modified as follows:

"Methylene blue has been proven helpful for surgeons during the learning phase of radiopharmaceutical-driven SLC dissection [45]. Nevertheless, it needs direct visualization, and it is an exclusively intraoperative technique [2, 19, 36], which may lead to multiple lymphocentrums or more aggressive tissue dissection to identify stained SLNs. Its association with transcutaneous mapping methods (i.e., radiopharmaceutical or NIR fluorescence with indocyanine green) helps to reduce those issues minimizing the surgical dose and the consequent risk of complications [2, 5, 19], [27, 46]. In human medicine, MB use is debated due to the possible side effects, which have not been reported in veterinary medicine [1, 2]. Also, in this study, the 9% local side effect observed is likely more correlated to the degranulation following injection than MB. The reported agreement in SLN detection between MB and NIR fluorescence or lymphoscintigraphy ranges from 51 to 100% [1, 2, 5, 24]. Lately, two articles on head and neck tumors reported the excision of a very low number of "blue" but "non-hot" or "non-fluorescent" SLNs [5,15], but all stained SLN were also reported positive for lymphoscintigraphy or NIR fluorescence in other studies [1, 2, 24]. In our sample, the concordance between blue staining and the radioactive count was 84%, similar to other reports in both veterinary and human literature [2, 5, 27, 47], reaching a statistically "fair" agreement [41]. Nonetheless, the high percentage (93%) of blue SLNs reported should be carefully interpreted as we always performed a combined exploration by applying the IGP as a first-line procedure. Therefore, the detection rate of the sole use of MB could be less than what was achieved with both techniques in the present paper.”

Lines 328-333: The use of methylene blue in Rossanese’s study is distinctly different to the use of methylene blue in this study. It is inappropriate to try to draw comparisons in rates of success here.

Authors: the Authors agree with REV2 and the paragraph has been removed.

Lines 334-335: Again, it should be specified whether this is 91% of dogs that had a node identified, or whether it allowed identification of 91% of nodes identified on pre-operative lymphoscintigraphy. Regardless, neither is the same as a blanket sensitivity percentage, as that would require additional exploration of regional nodes. 

Authors: In this sentence we are talking of SLN and not SLC. It implies that the percentage is related to the total number of SLN identified and removed within the total number of SLC dissected based on planar lymphoscintigraphy indication.

As previously explained, the preoperative planar lymphoscintigraphy cannot detect the number of SLN included in an SLC. Therefore, preoperative planar lymphoscintigraphy identifies only SLCs. In this study, the Authors did not compare performance to detect SLN between methylene blue and scintigraphy, but in the individual case in which the preoperative scintigraphy revealed the lymphocentrum, how many sentinel lymph node/s collected radioactivity and/or stained blue and how these correlates to the presence of lymph node metastasis according to Weishaar grades.

If there was no agreement between the regional and sentinel lymph nodes, we did not remove the regional lymph node. As shown in previous experimental studies on dogs, these techniques can identify correctly SLN (Aquino JU et al. Acta Cir Bras. 2012;27(2):102-8. doi: 10.1590/s0102;  Suami H et al. PLoS One. 2013;8(7):1-9.). In addition, a recent study shows that removing the regional lymph node and the sentinel lymph node is unnecessary to exclude a possible false negative. (Ferraris E et al. Vet Comp Oncol. 2023 Jan 12. doi: 10.1111/vco.12878)

We try to clarify it as follows: Now lines 440-441 “The IGP was extremely helpful during the detection of SLNs within the identified SLC, leading to a detection rate of 91%.”

Lines 355-357 (now 550-552): Please review this first sentence for phrasing.

Authors: the sentence was changed as follows. “In three separate SLC (one mandibular and two popliteal), only 1 non-hot and non-blue SLN was identified. In one of these SLC (popliteal), the non-hot and non-blue SLN bearing an early metastasis (HN2)

Lines 361-363: Absolutely, and yet this possibility is largely sidelined for most of the paper.

Authors: The Authors think the hypothesis reported in the discussion was calibrated on our inclusion criteria and results. In addition, these hypotheses are supported by references. To better clarify our replay to REV2, the Authors underline that 32 out of 196 SLN showed different staining: 10 blue – non-hot, 12 non-blue – hot and ten non-blue – non-hot. These were 16% of all SLN. Also, in all cases included, preoperative lymphoscintigraphy identified the lymphocentrum and occult overtly nodal metastasis were diagnosed in 20 out of 196 SLN. Considering that the study includes dogs with no palpable and normal size lymph nodes without clinical or ultrasonographic evidence or suspected RLN metastasis in which the pre-operative planar scintigraphy was able to identified a SLC, the hypothesis that extensive nodal metastasis could modified the lymphatic flow was not stressed in the discussion (lines 571-576).

Lines 399-404: A similar time for resection cannot be used as an indication that nodes were not fixed. It may well have been that no nodes were fixed but the time of resection is not an appropriate marker of this.

Authors: the sentence was modified as follows (lines 623-628)

At least in other cancer histotypes, metastatic SLNs can result enlarged, palpable, stiffer and more firmly anchored to the surrounding tissues [43, 56]. In our sample, we could not identify any adhesion to surrounding tissues or other clinical characteristics at admission and during surgery that may have suggested the metastatic status before histopathological evaluation.

Lines 419-423: The limitations section is insufficient based on the study design and should more thoroughly explore the limitations of a retrospective study, without any form of control or assessment for false negatives. 

Authors: Based on our replies to REV2 questions (general comments and specifics comments) in particular regarding regional lymph nodes (control group) the Authors think that the limitations described are adequate to aim, study design and results.

We modified the sentence at lines 649-651: ‘’Furthermore, the presence of false negatives for the mapping techniques described was not evaluated, however, the oncological outcome was beyond the aim of the present study, and only the lymph node staging was used’’

Reviewer 3 Report

This interesting study evaluated the combined use of MB and IGP after preoperative planar scintigraphy for SLN detection in dogs with MCT and also the association with the metastatic status of the removed SLNs.

I have some comments and doubts about the methodology, as well as some suggestions that could improve the quality of the article, as follows:

·      L112-121: It is not clear if the decision for removal of all LNs was influenced by the grade of the MCT, since cytological reports may contain this kind of information. Indeed, in high-grade MCT cases, the persistence of the surgeons in finding a higher number of LNs could influence the results.

·      L119-121: Why and how the volume and sites of injection of the tracers were reduced?

·      L122-123: please add a reference for this criterium “at least 10% of RC”. Could this threshold be influenced by the reduction of the tracers injected or by the size of the LNs? If yes, this should be pointed as a limitation of the study.

·      L123-125: This is not what Figure 2’s legend is telling us… In this case, only the hot LNs were removed.

·      L-135-138: So, in this case, A and B were not removed? What if they had metastases too? The numbers (and conclusions, possibly) would change, especially in Table 2 for the “non-hot” LNs…

·      L141-144: How many LN sections were evaluated for HN grading in each case? On H&E slides? Any special staining? Who evaluated the slides? Only one person, the same person? Experienced vet pathologist? I suggest re-evaluating all histological samples, by the same pathologist (more than one, ideally 3), and being blinded to the diagnoses and MCT grades.

·      L163-164: Information about Excel is unnecessary, I think.

·      L168-173: I am not sure if Mann-Whitney is the adequate test for all these comparisons.

·      L193: “French bulldog”

·      L197: Please inform SD and/or if this is average for weight

·      L208-212: Only 1 grade III and 2 high-grade? Did you review the original diagnoses? I recommend doing this with 3 independent and experienced pathologists, to minimize interobserver variabilities.

·      L234-235: Do all LNs have the same expected size? Did you compare by LN location?

·      L242-247: table 2 doesn’t show this.

·      L242-258: This paragraph is overall confusing. Since these data are in the table, I think you could just highlight the most important information and p values. Which statistical test did you use to compare the distributions?

·      L303-305: How many different surgeons? This should be stated and discussed as a limitation of the study.

·      L356: one of mandibular or among the popliteal?

·      L419-423: I suggest including the limitations commented above and also the use of the Weishaar classification. Particularly for MCTs, it is very difficult to be sure about metastases, since there are no objective established criteria to do this. HN3 LNs are very easy to detect and identify. A useful diagnostic method should increase the precision of detecting small metastases, HN2 and, possibly, HN1. Moreover, HN1 is an arbitrary criterion of the Weishaar classification, because the number of mast cells was not defined scientifically/statistically, as well as we are not sure that these cells are really neoplastic, resident mast cells, or even macrophages that have phagocytised granules…

·      L430-431: All the LNs should be removed… but figure 2 says that some were not.

Minor issues:

·      L75: SLC does

·      The LNs were not “metastatic”, they presented or contained tumour metastases.

·      “Hence” is repeated several times throughout the text

·      English language editing is needed.

Author Response

The Authors are grateful to Reviewers for their questions and suggestions that helped improving the paper. The Authors thank the Reviewers for the time dedicated to their paper.

REV3

This interesting study evaluated the combined use of MB and IGP after preoperative planar scintigraphy for SLN detection in dogs with MCT and also the association with the metastatic status of the removed SLNs.

I have some comments and doubts about the methodology, as well as some suggestions that could improve the quality of the article, as follows:

  • L112-121: It is not clear if the decision for removal of all LNs was influenced by the grade of the MCT, since cytological reports may contain this kind of information. Indeed, in high-grade MCT cases, the persistence of the surgeons in finding a higher number of LNs could influence the results.

Authors: In the present retrospective study, MCT cytological grading data were unavailable. 

However, at the moment, there was no available literature supporting that cytological information on the primary MCT would have influenced the decision to remove the lymph node. Therefore, pre- and intra-operative mapping was performed in all MCTs regardless of the other clinical and pathological features of the tumor (moreover, the histopathological features were available only after the removal of the tumor and lymph nodes) Ferrari et al 2021.

The number of lymph nodes removed per lymphocentrum was determined only by the intraoperative guidance of the mapping techniques: all SLNs identified in an SLC have been removed. The surgeon removed all lymph nodes encountered during sentinel lymphocentrum exploration (blue/non-blue hot/non-hot). The dissection continued until RC was less than 10% of the hottest SLN extirped, and no other lymph nodes were visible/palpable.

  • L119-121: Why and how the volume and sites of injection of the tracers were reduced?

Authors Line 119-121: In MCT, less than 0.5 cm in maximum diameter, performing a peritumoral injection in 4 quadrants with a volume of 0.4 ml of radiotracer was impossible. Performing four injections in such a small area will often cause bleeding and leaking of the injected radiotracer from the previously performed needle holes. In those cases, the volume of radiotracer was reduced (0.2 ml) and a single injection site technique. This approach was recently supported by the publication of Thai et al. (Thai JN, Shamis M, Gokli A, Demissie S, Landau E, Chaya N, Peti S, Brenner AI. Single shot lymphoscintigraphy in breast cancer: Effective single tracer sentinel node detection protocol with reduction in procedural pain. Clin Imaging. 2022 Apr;84:43-46. doi: 10.1016/j.clinimag.2022.01.015. Epub 2022 Feb 3. PMID: 35134675.) In dogs with smaller mast cell tumors, despite the reduction in injected volume, the activity administered was the same as in all other patients. We slightly modified the sentence (139-141).

  • L122-123: please add a reference for this criterium “at least 10% of RC”. Could this threshold be influenced by the reduction of the tracers injected or by the size of the LNs? If yes, this should be pointed as a limitation of the study.

Authors: This was the choice of the authors of the present study. The relation between the radioactive count of SLN and the tumor (site of injection of the 99mTc) was considered to determine if the node was ‘’hot’’. The choice was based on the relationship between nodal and injection point RC was the most standardizable and replicable on all patients.

The tracer reduction refers only to the volume and not to the injected activity, which was held constant. Therefore, this threshold cannot be affected by the reduction in injected volume or the size of the lymph node.

In addition the authors want to clarify that preoperative planar lymphoscintigraphy is not a morphologic technique, so it does not tell how many lymph nodes are within an SLC; indeed, it is a functional technique, i.e., it tells us that there is an uptake of the radiotracer.

  • L123-125: This is not what Figure 2’s legend is telling us… In this case, only the hot LNs were removed.

Authors: The Authors thank the Reviewer for the comment. Figure 2 showed a scenario where, despite hot lymph nodes (C-D) being more superficial than non-hot nodes, the surgeon, during the lymphocentrum exploration (using palpation or because nodes are visible), can recognize other lymph nodes not hot or hot or not blue that are located closer to hot SLN. The text in Figure 2 has been modified.

Figure 2. Intraoperative scenario of SLC exploration using IGP, with at least one “hot and blue” (C) or “hot and non-blue” (D) SLN less deep than other non-hot SLNs (A and B). The surgeon removes the “hot” lymph nodes encountered during sentinel lymphocentrum exploration and all other lymph nodes identified using palpation or because they are very close (A) or visible (B) to hot nodes (C-D). The surgeon evaluates the field RC at the end of the surgical exploration. The exploration is stopped when the RC is less than 10% of the hottest SLN extirped, and no other lymph nodes are visible/palpable.

  • L-135-138: So, in this case, A and B were not removed? What if they had metastases too? The numbers (and conclusions, possibly) would change, especially in Table 2 for the “non-hot” LNs…

Authors: As previously mentioned, the description of the figures has been modified to clarify the concept. All lymph nodes that are identified during lymphocentrum dissection are removed: blue, hot, non-blue and non-hot but visible and palpable.

  • L141-144: How many LN sections were evaluated for HN grading in each case? On H&E slides? Any special staining? Who evaluated the slides? Only one person, the same person? Experienced vet pathologist? I suggest re-evaluating all histological samples, by the same pathologist (more than one, ideally 3), and being blinded to the diagnoses and MCT grades.

Authors: The Authors thank the Reviewer for the observations. The required details have been added in the manuscript (lines 123-131)

Text (Materials and methods): Each lymph-node was trimmed following the same procedure: the lymph-node was divided in two halves with a longitudinal cut through the hilus. When the lymph-node was thicker than 3 mm (minor axis), additional parallel cuts were performed obtaining multiple slices (1.5 mm-thick each) from each half. The whole sample was processed and for each slice, two serial microtomic sections were cut and stained with haematoxylin-eosin and Giemsa stain, respectively. Histological evaluation of the excised SLN was performed according to Weishaar [6].

Lymph nodes and MCT grades were examined independently and blinded by three experienced pathologists and revised then collegially for the purpose of the study.

  • L163-164: Information about Excel is unnecessary, I think.

Authors: Removed.

  • L168-173: I am not sure if Mann-Whitney is the adequate test for all these comparisons.

Authors.: the Authors thank the Reviewer for the indication. Since all the variables were not-normally distributed, even if they might be regarded as discrete and not continuous, the Mann-Whitney’s test result the most fitting. (Statistics for Veterinary and Animal Science, 3rd Edition, Aviva Petrie, Paul Watson, Eds, 2013). If the Reviewer has some specific indication, the Authors would gladly accept his suggestion and apply the suggested test.

  • L193: “French bulldog”

Authors: Thanks for the comment. As requested by another reviewer, only the most representative breeds were reported, while less represented breeds, including the French bulldog, were removed.

  • L197: Please inform SD and/or if this is average for weight

Authors: the Authors thank the Reviewer for the indication. As stated in materials and methods, non-normally distributed variables were reported as median (range). The text was changed in “and had a median bodyweight of”. (lines 255-256)

  • L208-212: Only 1 grade III and 2 high-grade? Did you review the original diagnoses? I recommend doing this with 3 independent and experienced pathologists, to minimize interobserver variabilities.

Authors: Although it can be considered peculiar, few grade III and high grade MCTs were actually present in our cohort of cases. Three experienced pathologists, two of them board diplomates and authors of several papers concerning MCTs (in addition to papers focus on SLN, Stefanello et al  JVIM 2009, Stefanello et al JAVMA 2010), revised the results (see before) and were in complete agreement. This low number could be linked to the inclusion criteria that allowed the incorporation of dogs with clinically normal lymph node, probably excluding a portion of high-grade tumor in which lymph node metastasis is more prone to be clinically visible and diagnosed before surgery. The reason why in the results there is 1 grade III Patnaik and 2 high grade Kiupel is that one of the Kiupel high grade was defined as grade II Patnaik.

  • L234-235: Do all LNs have the same expected size? Did you compare by LN location?

Authors: The Authors did not estimate an expected nodal size, since to their knowledge, lymph node size has low correlation with body weight and age, as discussed in lines 613-622. Moreover, the Authors are not aware of any difference in lymph nodes size within a subject for different locations, and LN dimensions were not compared between locations. All dogs had normal-sized lymph nodes without clinical or ultrasonographic evidence of suspected loco-regional metastasis. Please see replies to general comment Rev 2.

  • L242-247: table 2 doesn’t show this.

Authors:  Changes were made to the tables and two more new tables were added as the request by REV1. Therefore, the text-to-table correspondence has now been changed based on this comment.

  • L242-258: This paragraph is overall confusing. Since these data are in the table, I think you could just highlight the most important information and p values. Which statistical test did you use to compare the distributions?

Authors: Thanks for the comment. As in response to the previous comment, the text and tables have now been edited to clarify the data. As reported in materials and methods, the chi-square test was applied to compare distributions, or better the proportions, i.e., the relative numbers of metastatic/non-metastatic within the described categories – see the specification added in the materials and methods section. “In particular, contingency tables were built up as follows: number of metastatic and non-metastatic SLCs vs. number of regional and sentinel lymphocentrums; number of metastatic and non-metastatic SLNs vs. number of regional and sentinel lymph nodes; number of metastatic and non-metastatic SLCs vs. number of lymphocentrums in each of the staining category (blue–hot, non-blue–hot, blue–non-hot, non-blue–non-hot); number of metastatic and non-metastatic SLNs vs. number of lymph nodes in each of the staining category (blue–hot, non-blue–hot, blue–non-hot, non-blue–non-hot); number of lymph nodes in each HN class vs. number of lymph nodes in each of the staining category (blue–hot, non-blue–hot, blue–non-hot, non-blue–non-hot).” (lines 221-230) (Statistics for Veterinary and Animal Science, 3rd Edition, Aviva Petrie, Paul Watson, Eds, 2013).

  • L303-305: How many different surgeons? This should be stated and discussed as a limitation of the study.

Authors: Considering the study's aims, the Authors do not fully understand the question. LN removal should be considered a generally easy surgical procedure (excluding sacral and iliac lymph nodes location), and in SLNs mapping and removal, the learning curve of the use of the gamma probe is more important rather than the surgical experience of the operator.

To answer REV3's question, as we stated in response to REV2, this project on the SLN started in 2017, and 3 surgeons with the same learning curve performed the surgeries.

To prove the experience, we can report the articles published by our group (please see comments to REV2). The surgeons are: Prof. Dr. Damiano Stefanello (DS), Dr. Roberta Ferrari (RF) and Dr. Lavinia Chiti (LC).

The specific statement at line 303-305 was removed as request by REV2, however the subjective evaluation of the lower disturbance of the blue-stained tissue in the primary tumour removal than what reported in the first paper (Ferrari et al, Vet Surg 2020) is actually linked to the progressive of the learning curve and to the increase number of cases.

  • L356: one of mandibular or among the popliteal?

Authors: Popliteal. It is now added in the text (562-564) as requested also by REV2

  • L419-423: I suggest including the limitations commented above and also the use of the Weishaar classification. Particularly for MCTs, it is very difficult to be sure about metastases, since there are no objective established criteria to do this. HN3 LNs are very easy to detect and identify. A useful diagnostic method should increase the precision of detecting small metastases, HN2 and, possibly, HN1. Moreover, HN1 is an arbitrary criterion of the Weishaar classification, because the number of mast cells was not defined scientifically/statistically, as well as we are not sure that these cells are really neoplastic, resident mast cells, or even macrophages that have phagocytised granules…

Authors: Even though the classification of Weishaar for lymph nodes has some limitations, it is the current grading system applied for the assessment of nodal metastases of mast cell tumors and its wide application since its publication in 2014 has produced a considerable amount of comparable data. Although a revision of the Weishaar classification is probably advisable, it is beyond the scope of the present study.

As far as the histological identification of mast cells is concerned, it is the authors’ opinion that even though the intracytoplasmic granules of macrophages can be (although not necessarily or always are) slightly positive for Giemsa staining, they are morphologically distinct since, as opposite to mast cells granules, they are larger and not uniform. In addition, the cell tends to be larger, and the nucleus is more likely to be peripheral in macrophages than mast cells.

  • L430-431: All the LNs should be removed… but figure 2 says that some were not.

Authors: The text of the figure was changed in according to your previous comment.

Minor issues:

  • L75: SLC does

Authors: Done.

  • The LNs were not “metastatic”, they presented or contained tumour metastases.

Authors: We understand the meaning comment of the Rev3, but substitution of this term in the text could overburn some sentences, so it have remained unchanged. Furthermore, the term ‘’metastatic’’ is reported in literature referred to lymph nodes (i.e. Beer et al 2022; Guerra et al 2022; Ferrari et al 2020).

  • “Hence” is repeated several times throughout the text

Authors: Done. It has now been replaced in the text by synonyms.

  • English language editing is needed.

Authors: Done.

Round 2

Reviewer 2 Report

Thank you for your comments and the time taken for revision. I believe that the majority of the points where we have not seen eye to eye have resulted from differing perspectives regarding the real or inadvertently implied scope of the study. With the previously stated aims of describing “the results of the combined use of MB and IGP, after preoperative planar scintigraphy, for SLN detection in dogs with MCT, in a 5 years single-center experience”, phrasing such as “the association with MB allowed the identification of 98% of metastatic SLNs” implies to me a more general assessment than the work that has been done. Given the authors’ clarifications, I feel that more specific aims and greater specificity in context when reporting aspects such as nodal detection rate would limit the potential for misinterpretation. I have included comments to this effect below. 

I do not want to presume to change the wording of someone else’s title myself, but I would also strongly recommend a shift to a title that uses more specific language than “usefulness”. As usefulness is not defined, it might be readily taken to have broader implications than assessed here. How clinically useful is any of this to these dogs? I suspect it is, but without data about impact on their outcomes we can’t say. Greater specificity of language can avoid this morass. I hope this makes sense.

The authors’ characterisation of the Ferraris paper as indicating it is unnecessary to assess other nodes is a touch simplistic. First, this manuscript evaluated a different technique. Differing techniques may provide differing results, as demonstrated most recently in Sanchez-Alvarez, et al. Vet Surg, 2023. In addition, of the 23 negative SLNs in the Ferraris study, 3 (13%) had a positive non-sentinel node. While this is not a high proportion and may be considered clinically acceptable, we should not ignore the potential for failure and assume validity in every case, as that will not provide an accurate reflection of the technique(s) assessed. For example, if pre-op planar lymphoscintigraphy also had a 13% false negative rate, the ability of IGP and methylene blue to identify 98% of lymph nodes called sentinel on imaging would correspond to identification of approximately 85% of affected nodes. I also do not feel that the Aquino and Suami papers provide sufficiently robust assessment to assume the validity of every sentinel lymph node technique at every anatomic site. It is unreasonable to expect a study to be able to answer every question related to a topic and I apologise if that is how my comments came across; however, it is important to acknowledge limitations and provide sufficiently specific descriptions. This study can and does assess agreement between methylene blue and IGP. This can be useful information. 

Thank you for providing information regarding numbers of dogs included in other studies. While I agree that it is reasonable to include individual cases in multiple studies, I strongly disagree, however, that declaration of inclusion in multiple, related studies, is unnecessary. Even if the aims are slightly different, each could have overlap in results with others, such as in reporting of metastatic rate amongst nodes assessed. A researcher looking at systematic review could otherwise conclude that this is a body of work in 374 dogs, whereas the real number is closer to a third of that. Case numbers are frequently a limitation in veterinary research but acknowledgment of overlap in populations allows contextualisation for readers and can limit over-extrapolation without invalidating the work. In addition, the process of enrolment and data accrual is different for a prospective study than a purely retrospective study. There are some potential benefits, including that the quality of data is likely better. The authors and their group have clearly worked hard to generate clinical data in a clinically relevant field, and I do not think citing their own work in this context is inappropriate. Presenting this dataset as an isolated, retrospective population is misleading though (even if unintentionally so) and I cannot support publication without an acknowledgement of overlap. I have included a suggestion for how to do so below. If the editor prefers another method, I would of course, defer to them. I would also remove characterisations of this study as retrospective and instead simply refer to it as a cross-sectional study.

Lines 12-24: The fact that all dogs were required to undergo pre-operative lymphoscintigraphy is not mentioned in either the simple summary or abstract. The pre-operative determination of sentinel lymph centers is a key aspect of the methodology. In addition, the stated aims of describing the use of methylene blue do not clearly indicate the intended comparison that the authors touch on in the main aims but describe more clearly in their reply (“The study intended to assess how MB added to IGP may increase detection rate of SLN, associated also to the staging value of this increased number of SLN extirpated”). I would suggest amending lines 14-15 to “This study was performed in dogs with mast cell tumours after preoperative assessment identification of draining nodes using scintigraphy (radiation). The aim was to assess whether the addition of methylene blue dye to intraoperative detection of radiation in lymph nodes increased surgical identification of nodes considered sentinel based on preoperative imaging.” Descriptions of detailed techniques for a lay audience are tough and the authors may be able to improve on this, but this phrasing should fit within the word limit and conveys 1) the requirement for pre-op scintigraphy and 2) that pre-op scintigraphic assessment is used as the standard. 

Lines 27-30: The use of methylene blue has been described in combination with IGP by your group, with dogs in this dataset. As such, I would suggest removing this as the first aim and consolidating to a single aim, in line with the comments in your reply, e.g. “The aim of this study was to assess whether MB increased surgical detection of SLN beyond use of IGP alone in clinically node-negative dogs with mast cell tumors, following detection of sentinel lymph nodes via planar lymphoscintigraphy.” This aim is more specific and justifiable as publishable data than description of an already described technique. 

Lines 88-92: If you’re ok with the suggested aim above, consider amending this to be consistent with the abstract. 

Lines 97-98: As this is not a typical retrospective study, I suggest amending this to “…from January 2017 to November 2022 were included in this cross-sectional study, which included dogs enrolled in multiple prospective studies. Details of overlap are included in the acknowledgements section.”

Line 157: If accurate, consider “The following information about the SLCs were recorded at the time of planar lymphoscintigraphy: …” This can convey that these data were not simply inferred from potentially variable records, given the previous description of a retrospective study. 

Lines 300-302: Consider revising to “Of the total number of lymph nodes biopsied based on planar lymphoscintigraphy, the IGP alone…”

Line 306: Amend to “…the IGP alone would have identified 91% of SLNs within the examined SLCs, whereas…”

Lines 313-315: Thank you for your reply. I am aware that lymphoscintigraphy is a widely used technique; it also has acknowledged false negatives in people. My issue with the phrasing was that without specifically referencing that detection was relative to pre-operative lymphoscintigraphy, this could be interpreted as indicating that these techniques identify 95% of all metastatic lymph nodes. I would suggest the phrasing “In the present study, the intraoperative combined use of both MB and IGP allowed for the detection of 95% of the SLNs considered sentinel based on preoperative planar lymphoscintigraphy, increasing the detection rate beyond IGP alone.”

Lines 315-317: I would suggest revising to “Moreover, considering only the portion of metastatic SLNs detected within the examined SLCs, the addition…”

Lines 340-341: For greater specificity, I would suggest revising to “The IGP was extremely helpful during the detection of SLNs within the SLC identified by planar lymphoscintigraphy, leading to detection of 91% of these nodes.”

Line 434: Consider “quasi-retrospective”

Author Response

REV 2

Comments and Suggestions for Authors

Thank you for your comments and the time taken for revision. I believe that the majority of the points where we have not seen eye to eye have resulted from differing perspectives regarding the real or inadvertently implied scope of the study. With the previously stated aims of describing “the results of the combined use of MB and IGP, after preoperative planar scintigraphy, for SLN detection in dogs with MCT, in a 5 years single-center experience”, phrasing such as “the association with MB allowed the identification of 98% of metastatic SLNs” implies to me a more general assessment than the work that has been done. Given the authors’ clarifications, I feel that more specific aims and greater specificity in context when reporting aspects such as nodal detection rate would limit the potential for misinterpretation. I have included comments to this effect below. 

Authors: we thank the Reviewer for clarifying his/her point of view, and we made every effort to follow his/her indication.

I do not want to presume to change the wording of someone else’s title myself, but I would also strongly recommend a shift to a title that uses more specific language than “usefulness”. As usefulness is not defined, it might be readily taken to have broader implications than assessed here. How clinically useful is any of this to these dogs? I suspect it is, but without data about impact on their outcomes we can’t say. Greater specificity of language can avoid this morass. I hope this makes sense.

Authors: we changed the title as follow:

Evaluation of surgical aid of methylene blue in addition to intraoperative gamma probe for sentinel lymph nodes extirpation in 116 canine mast cell tumors (2017-2022)

The authors’ characterisation of the Ferraris paper as indicating it is unnecessary to assess other nodes is a touch simplistic. First, this manuscript evaluated a different technique. Differing techniques may provide differing results, as demonstrated most recently in Sanchez-Alvarez, et al. Vet Surg, 2023. In addition, of the 23 negative SLNs in the Ferraris study, 3 (13%) had a positive non-sentinel node. While this is not a high proportion and may be considered clinically acceptable, we should not ignore the potential for failure and assume validity in every case, as that will not provide an accurate reflection of the technique(s) assessed. For example, if pre-op planar lymphoscintigraphy also had a 13% false negative rate, the ability of IGP and methylene blue to identify 98% of lymph nodes called sentinel on imaging would correspond to identification of approximately 85% of affected nodes. I also do not feel that the Aquino and Suami papers provide sufficiently robust assessment to assume the validity of every sentinel lymph node technique at every anatomic site. It is unreasonable to expect a study to be able to answer every question related to a topic and I apologise if that is how my comments came across; however, it is important to acknowledge limitations and provide sufficiently specific descriptions. This study can and does assess agreement between methylene blue and IGP. This can be useful information. 

Authors: we thank the reviewer for the comment. We erased the added paragraph (lines 455-456) and the reference to Ferraris et al. 2023. Moreover, we added the limitation requested, discussing it (lines 469-475 file word): Secondly, we did not collect and analyzed regional lymph nodes. This might have decreased the ability to exclude regional involvement. It should be carefully considered that the SLN is by definition the first lymph node draining a primary tumor, which harbors a higher probability for metastatic seeding. Nonetheless, it has been described that lymph nodes might be involved via the connecting tumor-associated lymphatic vasculature, over non-sentinel lymph nodes within the tumor-draining basin and distant uninvolved lymph nodes [60].

Thank you for providing information regarding numbers of dogs included in other studies. While I agree that it is reasonable to include individual cases in multiple studies, I strongly disagree, however, that declaration of inclusion in multiple, related studies, is unnecessary. Even if the aims are slightly different, each could have overlap in results with others, such as in reporting of metastatic rate amongst nodes assessed. A researcher looking at systematic review could otherwise conclude that this is a body of work in 374 dogs, whereas the real number is closer to a third of that. Case numbers are frequently a limitation in veterinary research but acknowledgment of overlap in populations allows contextualisation for readers and can limit over-extrapolation without invalidating the work. In addition, the process of enrolment and data accrual is different for a prospective study than a purely retrospective study. There are some potential benefits, including that the quality of data is likely better. The authors and their group have clearly worked hard to generate clinical data in a clinically relevant field, and I do not think citing their own work in this context is inappropriate. Presenting this dataset as an isolated, retrospective population is misleading though (even if unintentionally so) and I cannot support publication without an acknowledgement of overlap. I have included a suggestion for how to do so below. If the editor prefers another method, I would of course, defer to them. I would also remove characterisations of this study as retrospective and instead simply refer to it as a cross-sectional study.

Authors: we agree with the indication of the Reviewer, and added the following sentence in the Acknowledgements, as suggested: “Among the 103 canine patients included in the present study, 98 overlapped with presented cases in previous publications, with different aims, from our research group …” (lines 513-538)

Lines 12-24: The fact that all dogs were required to undergo pre-operative lymphoscintigraphy is not mentioned in either the simple summary or abstract. The pre-operative determination of sentinel lymph centers is a key aspect of the methodology. In addition, the stated aims of describing the use of methylene blue do not clearly indicate the intended comparison that the authors touch on in the main aims but describe more clearly in their reply (“The study intended to assess how MB added to IGP may increase detection rate of SLN, associated also to the staging value of this increased number of SLN extirpated”). I would suggest amending lines 14-15 to “This study was performed in dogs with mast cell tumours after preoperative assessment identification of draining nodes using scintigraphy (radiation). The aim was to assess whether the addition of methylene blue dye to intraoperative detection of radiation in lymph nodes increased surgical identification of nodes considered sentinel based on preoperative imaging.” Descriptions of detailed techniques for a lay audience are tough and the authors may be able to improve on this, but this phrasing should fit within the word limit and conveys 1) the requirement for pre-op scintigraphy and 2) that pre-op scintigraphic assessment is used as the standard. 

Authors: we thank the Reviewer for her/his suggestion and we changed the abstract including the following sentence: “This study was performed in dogs with mast cell tumors after preoperative assessment identification of draining nodes using lymphoscintigraphy. The aim was to assess whether the addition of methylene blue dye to intraoperative detection of radioactivity in lymph nodes increased surgical identification of such nodes.” (lines 14-17).

Lines 27-30: The use of methylene blue has been described in combination with IGP by your group, with dogs in this dataset. As such, I would suggest removing this as the first aim and consolidating to a single aim, in line with the comments in your reply, e.g. “The aim of this study was to assess whether MB increased surgical detection of SLN beyond use of IGP alone in clinically node-negative dogs with mast cell tumors, following detection of sentinel lymph nodes via planar lymphoscintigraphy.” This aim is more specific and justifiable as publishable data than description of an already described technique. 

Authors: we thank the Reviewer for her/his suggestion and we changed the abstract including the following sentence: “The aim of this study was to assess whether MB increased surgical detection of SLN beyond use of intraoperative gamma-probe (IGP) alone in clinically node-negative dogs with mast cell tumors (MCTs), following detection of sentinel lymphocentrums (SLCs) via preoperative planar lymphoscintigraphy.” (lines 27-30).

Lines 88-92: If you’re ok with the suggested aim above, consider amending this to be consistent with the abstract. 

Authors: we thank the Reviewer for her/his suggestion and we deleted the secondary aim according to the abstract. (lines 110).

Lines 97-98: As this is not a typical retrospective study, I suggest amending this to “…from January 2017 to November 2022 were included in this cross-sectional study, which included dogs enrolled in multiple prospective studies. Details of overlap are included in the acknowledgements section.”

Authors: we thank the Reviewer for her/his suggestion and we change the phrases accordingly

“, which included dogs enrolled in multiple prospective studies. Details of overlap are included in the acknowledgements section.” (lines 117-118)

Line 157: If accurate, consider “The following information about the SLCs were recorded at the time of planar lymphoscintigraphy: …” This can convey that these data were not simply inferred from potentially variable records, given the previous description of a retrospective study. 

Authors: we thank the Reviewer for her/his suggestion and we change the phrases accordingly

(lines 183-184)

Lines 300-302: Consider revising to “Of the total number of lymph nodes biopsied based on planar lymphoscintigraphy, the IGP alone…”

Authors: we thank the Reviewer for her/his suggestion and we change the phrases accordingly

“Of the total number of lymph nodes biopsied within a SLC as identified by preoperative planar lymphoscintigraphy, the IGP alone…” (lines 328-329)

Line 306: Amend to “…the IGP alone would have identified 91% of SLNs within the examined SLCs, whereas…”

Authors: we thank the Reviewer for her/his suggestion and we change the phrases accordingly

(line 335)

Lines 313-315: Thank you for your reply. I am aware that lymphoscintigraphy is a widely used technique; it also has acknowledged false negatives in people. My issue with the phrasing was that without specifically referencing that detection was relative to pre-operative lymphoscintigraphy, this could be interpreted as indicating that these techniques identify 95% of all metastatic lymph nodes. I would suggest the phrasing “In the present study, the intraoperative combined use of both MB and IGP allowed for the detection of 95% of the SLNs considered sentinel based on preoperative planar lymphoscintigraphy, increasing the detection rate beyond IGP alone.”

Authors: we thank the Reviewer for her/his suggestion and we change the phrases accordingly

“In the present study, the intraoperative combined use of both MB and IGP allowed for the detection of 95% of the SLNs considered sentinel based on preoperative planar lymphoscintigraphy, increasing the detection rate beyond IGP alone.” (lines 345-346)

Lines 315-317: I would suggest revising to “Moreover, considering only the portion of metastatic SLNs detected within the examined SLCs, the addition…”

Authors: we thank the Reviewer for her/his suggestion and we change the phrases accordingly

“Moreover, considering only the portion of metastatic SLNs detected within the examined SLCs, the addition…” (line 347)

Lines 340-341: For greater specificity, I would suggest revising to “The IGP was extremely helpful during the detection of SLNs within the SLC identified by planar lymphoscintigraphy, leading to detection of 91% of these nodes.”

Authors: we thank the Reviewer for her/his suggestion and we change the phrases accordingly

“The IGP was extremely helpful during the detection of SLNs within the SLC identified by planar lymphoscintigraphy, leading to detection of 91% of these nodes.” (line 376)

Line 434: Consider “quasi-retrospective”

Authors: we thank the Reviewer for her/his suggestion; we prefer to avoid the term “retrospective” at all in this sentences (line 466).

Reviewer 3 Report

Thank you for the opportunity to discuss your results. Congratulations!

Author Response

Thank you.